# Role of *miR-944/MMP10/AXL*- axis in lymph node metastasis in tongue cancer

Bhasker Dharavath [1,2], Ashwin Butle[1], Ankita Pal[1], Sanket Desai[1,2], Pawan Upadhyay[1,2], Aishwarya Rane[1], Risha Khandelwal[1], Sujith Manavalan [1], Rahul Thorat[3], Kavita Sonawane[4], Richa Vaish[2,4], Poonam Gera[2,5], Munita Bal[2,6], Anil K. D'Cruz[4,7], Sudhir Nair [2,4✉] & Amit Dutt [1,2✉]

Occult lymph-node metastasis is a crucial predictor of tongue cancer mortality, with an unmet need to understand the underlying mechanism. Our immunohistochemical and real-time PCR analysis of 208 tongue tumors show overexpression of Matrix Metalloproteinase, MMP10, in 86% of node-positive tongue tumors ($n = 79$; $p < 0.00001$). Additionally, global profiling for non-coding RNAs associated with node-positive tumors reveals that of the 11 significantly de-regulated miRNAs, *miR-944* negatively regulates *MMP10* by targeting its 3'-UTR. We demonstrate that proliferation, migration, and invasion of tongue cancer cells are suppressed by *MMP10* knockdown or *miR-944* overexpression. Further, we show that depletion of *MMP10* prevents nodal metastases using an orthotopic tongue cancer mice model. In contrast, overexpression of *MMP10* leads to opposite effects upregulating epithelial-mesenchymal-transition, mediated by a tyrosine kinase gene, *AXL*, to promote nodal and distant metastasis in vivo. Strikingly, *AXL* expression is essential and sufficient to mediate the functional consequence of *MMP10* overexpression. Consistent with our findings, TCGA-HNSC data suggests overexpression of *MMP10* or *AXL* positively correlates with poor survival of the patients. In conclusion, our results establish that the *miR-944/MMP10/AXL*- axis underlies lymph node metastases with potential therapeutic intervention and prediction of nodal metastases in tongue cancer patients.

[1] Integrated Cancer Genomics Laboratory, Advanced Centre for Treatment, Research, and Education in Cancer, Kharghar, Navi Mumbai, Maharashtra 410210, India. [2] Homi Bhabha National Institute, Training School Complex, Anushakti Nagar, Mumbai, Maharashtra 400094, India. [3] Laboratory Animal Facility, Advanced Centre for Treatment, Research and Education in Cancer, Kharghar, Navi Mumbai, Maharashtra 410210, India. [4] Division of Head and Neck Oncology, Department of Surgical Oncology, Tata Memorial Hospital, Tata Memorial Centre, Parel, Mumbai 400012, India. [5] Tissue Biorepository, Advanced Centre for Treatment Research and Education in Cancer, Kharghar, Navi Mumbai, Maharashtra 410210, India. [6] Department of Pathology, Tata Memorial Hospital, Tata Memorial Centre, Parel, Mumbai 400012, India. [7] Apollo Cancer Center, Apollo Hospitals, CBD Belapur, Navi Mumbai 400614, India. ✉email: sudhirvr@gmail.com; adutt@actrec.gov.in

Oral cancer constitutes cancers of the mucosal surfaces of the lips, the floor of the mouth, oral tongue, buccal mucosa, lower and upper gingiva, hard palate, and retromolar trigone that are primarily associated with tobacco usage[1,2]. Among sub-sites, tongue cancer is the most predominant sub-site for intraoral cancer in developed countries with a varying incidence in developing countries[3]. An occult or subclinical metastasis to the lymph nodes close to the site of the primary tongue tumor— not detectable by imaging methods or physical examination— is associated with dismal prognosis in 27–40% of patients[4–6]. The single treatment modality of early stage oral cancer with a curative intent includes surgical resection of the primary tumor along with neck dissection with or without adjuvant therapy[7]. However, due to the presence of even a single cancer-positive lymph node, the loco-regional metastasis, as observed in one-third of oral cancer patients leads to a poor 5-year survival of below 50%[8].

The migration and invasion of cancer cells that underlie metastasis are driven by a plural set of intricate genotypic, phenotypic and micro-environmental processes. Proteins that are related to microvascular angiogenesis and lymphangiogenesis have been identified to promote regional lymph node metastases (LNM)[9]. A recent gene expression study, performed with advanced stage samples, reported the association of LNM to chromosomal instability and DNA repair defects[10]. We earlier reported significant upregulation of Matrix Metalloproteinase-10 (MMP10) in early stage tongue tumor samples associated with nodal metastases[11]. Overexpression of MMP10 has been reported to promote invasion, metastasis and regulate stemness of cancer cells through activation of Wnt signaling pathway in head and neck, and ovarian cancer, and promote tumor progression by regulating angiogenic and apoptotic pathways in cervical cancer[12–14]. Studies have demonstrated that MMP10 is required for transformed growth and invasion of non-small cell lung cancer cells in vitro[15], is induced in bronchioalveolar stem cells (BASCs) transformed by oncogenic KRAS[16], and promotes KRAS-mediated lung tumorigenesis in vivo[17]. Additionally, MMP10 has been reported to promote pre-metastatic niche formation[18] and found to be upregulated in early stage esophageal cancer patients[19]. However, the role of MMP10 remains unexplored in lymph node metastasis in early stage tongue cancer. Here, we validate that MMP10 overexpression in a cohort of 208 tongue tumor samples (including samples of sufficient quality from the N-zero clinical trial) is significantly correlated with nodal metastases. This study demonstrates that overexpression of MMP10 is sufficient and essential to induce tumor growth and nodal metastasis in an orthotopic mouse model. We also show that miR-944 negatively regulates MMP10, and AXL signaling pathway mediates phenotypes associated with MMP10 overexpressing tongue cancer. Furthermore, ectopic expression of miR-944 or knockdown of AXL suppresses MMP10-promoted proliferation, invasion, and migration of tongue cancer cells, indicating a therapeutic approach to target tongue cancer.

## Results

We previously reported over-expression of MMP10 and its association with nodal metastases in early tongue cancer patients[11]. In the current study, we validate MMP10 expression in a large cohort of tongue patient samples with nodal metastases and describe the mechanistic role of miR944/MMP10/AXL- axis in tongue cancer using in vitro biochemical and cell-based assays and in vivo orthotopic tongue tumor mouse model.

**MMP10 is upregulated in tongue cancer patients with lymph node metastasis.** To validate if MMP10 expression could stratify

tongue cancer patients likely to develop metastases, we performed immunohistochemistry (IHC) for MMP10 protein using 98 retrospectively collected clinical samples that were part of the phase-3 N-zero clinical trial (NCT00193765). On the basis of 500 patients with early stage (T1 and T2) oral cancer, the N-zero clinical trial compared the survival advantages of elective neck dissection (END) and therapeutic neck dissection (TND). The study established END as a standard of care for patients with early stage clinically nodal negative (T1, T2 and N0) oral cancer and showed its superiority in terms of overall and disease-free survival rates. The specificity of the MMP10 antibody was confirmed with the inclusion of primary antibody alone and secondary antibody alone negative controls (Supplementary Fig. 1a, b). MMP10 protein was mainly detected in the cytoplasm of tumor cells (Fig. 1d). IHC results suggest significant overexpression of MMP10 protein in tongue cancer patients with lymph node metastasis ($p = 0.0035$) (Fig. 1a, c and Supplementary Fig. 1c). Of note, an increased expression of MMP10 at the invasive fronts of the tumor was observed compared to the center of the tumor (Fig. 1d). Additionally, we screened for the expression of MMP10 transcripts by real-time PCR across 110 tongue tumor samples. The data suggest overexpression of MMP10 transcript in lymph node metastatic patient samples compared to the non-metastatic patient's group, ($p < 0.0001$) (Fig. 1b). In the clinical settings, given that both IHC and real-time PCR are routinely performed, our analysis suggests overexpression of MMP10 in 68/79 (86%) of patients with lymph

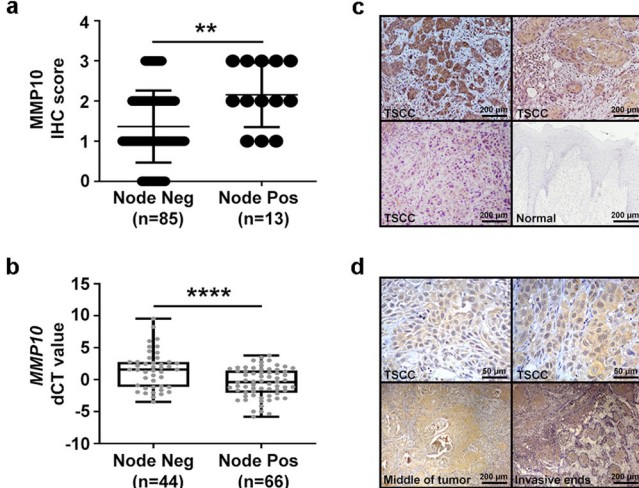

**Fig. 1 MMP10 overexpression is associated with nodal metastasis in tongue cancer patients. a** Immunohistochemistry of MMP10 in primary tongue tumor samples ($n = 98$). One dot represents IHC score of MMP10 in one sample. "Node Pos" and "Node Neg" represent primary tumor sample collected from patients with and without lymph node metastasis, respectively. Data are shown as means ± SD. **b** Quantitative real-time PCR (qRT-PCR) of MMP10 transcript expression in primary tongue tumor samples ($n = 110$). GAPDH was used as reference control. Data is plotted as boxplot representation of delta Ct (dCT) values. The middle line in the boxplot shows median along with the lower (Q1) and upper quartiles (Q3) as boxes. The whiskers represent the minimum and maximum values. **c** Representative IHC stained photomicrographs of tongue tumors with scale bar (200 μm). The brown color indicates positive staining for MMP10 in tumor samples with different staining intensities (strong, moderate, weak). The normal epithelium is negative for MMP10. **d** Representative IHC stained images showing expression of MMP10 in cytoplasm of tumor cells (scale bar = 50 μm), in the middle region of tumor and at the invasive ends (scale bar = 200 μm). p-values are from Mann–Whitney tests and denoted as **$p < 0.01$; ****$p < 0.0001$. Data shown in (b) are representative of $n = 3$ independent replicates for each sample.

**Table 1 Association of lymph node metastasis with *MMP10* expression in TCGA-tongue cancer data.**

| Samples | Nodal status | n (% along column) | *MMP10* expression, n (% along row) | | p-value |
|---|---|---|---|---|---|
| | | | Low expression | High expression | |
| TCGA-tongue cancer (n = 112) | Node positive | 64 (57%) | 44 (69%) | 20 (31%) | 0.0778 |
| | Node negative | 48 (43%) | 40 (83%) | 8 (17%) | |
| TCGA-early stage (T1-T2) tongue cancer (n = 62) | Node positive | 31 (50%) | 20 (65%) | 11 (35%) | 0.0816 |
| | Node negative | 31 (50%) | 26 (84%) | 5 (16%) | |

node metastasis, whereas only 11/79 (14%) of patients with lymph node metastasis has low expression of MMP10 (chi-square value = 33.53, p-value < 0.00001). Specifically, real-time PCR and IHC data reveal MMP10 overexpression in 58/66 (88%) and 10/13 (77%), and low expression in 8/66 (12%) and 3/13 (23%) of patients with lymph node metastasis, respectively. Overall, of 208 tongue tumors we find MMP10 overexpression in 86% of node-positive tongue tumors (n = 79; p < 0.00001). The findings suggest MMP10 is strongly correlated with nodal metastases in tongue cancer. Next, we correlated MMP10 expression with clinical parameters including age, gender, smoking, alcohol consumption, and tobacco consumption. However, no significant correlation was found with any of the mentioned clinical parameters (Supplementary Table 1). Furthermore, we correlated *MMP10* transcript expression with nodal metastases in the TCGA-tongue cancer data of early stage as well as early and advanced stage samples. The samples in both the comparisons (early stage and all stages of tongue cancer) were divided into quartiles independently based on FPKM values of *MMP10* and the upper quartile of the samples were considered as the samples with high expression of *MMP10* and the rest of the three quartiles were considered as the samples with low expression of *MMP10*. Interestingly, we found a marginally significant correlation (p < 0.082) of *MMP10* transcript expression with lymph node metastasis in both analyses (Table 1). Overall, we observed significant upregulation of *MMP10* in tongue cancer patients with nodal metastasis, across 208 in-house and 112 TCGA tongue cancer samples.

**MMP10 overexpression is essential and sufficient for cell proliferation, migration, and invasion of tongue cancer cells.** To investigate the role of *MMP10* in metastasis of tongue cancer, we screened for MMP10 expression in three tongue cancer cell lines (AW13516, AW8507, and CAL27) by quantitative real-time PCR and Western blotting. AW8507 and CAL27 cell lines express higher levels of *MMP10* compared to AW13516 cells at both transcript and protein levels (Supplementary Fig. 2a–c). The AW13516 cell line was used for overexpression of *MMP10*, and AW8507 and CAL27 cell lines were used for knockdown of *MMP10*. Stable clones of AW13516 overexpressing *MMP10* cDNA were generated and confirmed for MMP10 overexpression by real-time PCR and Western blotting (Fig. 2a). To investigate the role of *MMP10* in metastasis, invasion and migration assays were performed. In vitro, wound-healing assay results suggested a significant increase in the migratory potential of AW13516 cells upon overexpression of *MMP10* as compared to the AW13516-vector control cells (Fig. 2b). Further, invasion assay also suggested a significant increase in the invasive potential of AW13516 upon overexpression of *MMP10* (Fig. 2c), suggesting sufficiency of *MMP10* overexpression to induce cellular migration. In a reciprocal approach, to study if *MMP10* expression is essential for the metastatic properties of tongue cancer cells, shRNA-mediated knockdown of *MMP10* was performed in AW8507 and CAL27 cells. *MMP10* was stably downregulated in both cell lines using three different shRNAs (sh-1, sh-2, and sh-3) and compared with non-targeting control shRNA (sh-NT). Knockdown of MMP10

was confirmed at transcript and protein level by real-time PCR and Western blotting assay, respectively (Fig. 2d, h). Cell proliferation assay performed with the clones suggested a significant decrease (p < 0.001) in the proliferation rate of cells upon knockdown of *MMP10* (Fig. 2e, i). To check if difference in proliferation rate is due to the difference in viability or proliferation or apoptosis. Using propidium iodide (PI) staining and flow cytometer-based sorting of live and dead cells, we conducted a cell viability study with shRNA-mediated *MMP10* knockdown clones of AW8507 and CAL27. Results did not show significant difference in the viability of cells upon *MMP10* knockdown (Supplementary Fig. 3a, e). The results were further supported by immunoblotting, which revealed comparable amounts of caspase 3 and PARP cleavage in the *MMP10* knockdown clones of AW8507 and CAL27 compared to control cells (Supplementary Fig. 3a, e). We performed siRNA-mediated knockdown of *MMP10* in AW8507 and CAL27 cells. The knockdown of *MMP10* was confirmed using real-time PCR (Supplementary Fig. 3b, f). Similar to viability assay, Annexin V-FITC/PI staining did not show significant difference in apoptosis of cells upon knockdown of *MMP10* (Supplementary Fig. 3c, d, g, h). These findings demonstrate that the difference in cell proliferation rates following *MMP10* knockdown is what causes the reduction in cell proliferation rates. Furthermore, depletion of *MMP10* in both the cell lines led to a significant decrease (p < 0.01) in the migratory and invasive potential of cells (Fig. 2f, g, j, k). Taken together, these results suggest that upregulation of *MMP10* is sufficient and essential to promote migration and invasion of tongue cancer cells.

**Differentially expressed *miR-944* in primary tongue tumors targets 3'-UTR of *MMP10* to regulate lymph node metastasis.** We performed an in silico analysis to investigate miRNAs predicted to bind to 3'-UTR of *MMP10* gene using 10 different miRNA binding site prediction tools, as described in the Methods section. The analysis identified 7 candidate miRNAs (*miR-944, miR-496, miR-152-3p, miR-130a, miR-148a, miR-148b,* and *miR-453-3p*) targeting the 3'-UTR of *MMP10* by at least 7 of 10 prediction algorithms (Supplementary Table 2 and Supplementary Fig. 4b). The 7 candidate miRNAs were validated by real-time PCR across 20 primary tongue tumors and paired adjacent normal samples derived from patients with nodal metastases (Supplementary Fig. 4e, 5). A Pearson's correlation analysis between the delta-delta CT values revealed a significant negative correlation between *MMP10* transcript and *miR-944* of the 7 candidate miRNAs (Supplementary Fig. 4d), downregulated in lymph node-positive tumors (Supplementary Fig. 4c). Interestingly, *miR-944* was also found to be significantly downregulated (fold change < 0.5 and p-value ≤ 0.05) in tumors with nodal metastases based on global miRNA profiling using Affymetrix miRNA 3.0 GeneChip constituting 5,778 unique mature human microRNAs probe sets (Supplementary Table 3 and Supplementary Fig. 4a).

To confirm biochemically whether *miR-944* directly targets the 3'-UTR of *MMP10*, 3'-UTR of *MMP10* gene was cloned downstream of the luciferase open reading frame (ORF) to construct a

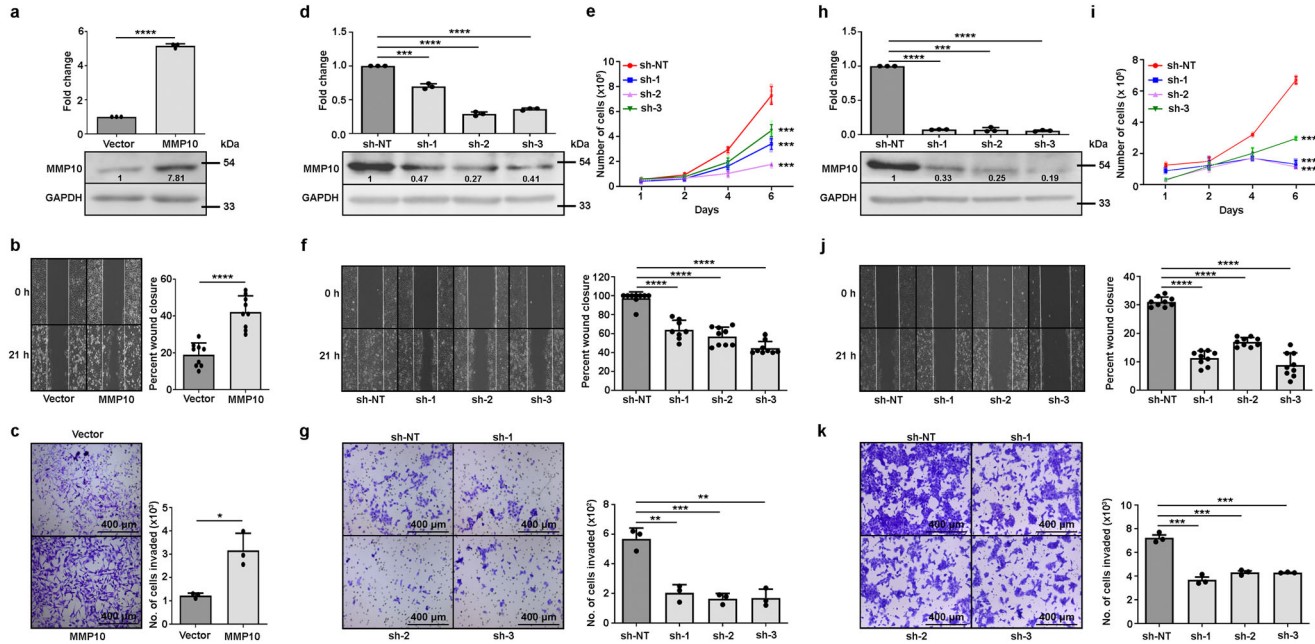

**Fig. 2 Genetic perturbation of *MMP10* affects cell proliferation, migration and invasion of tongue cancer cells. a** qRT-PCR and immunoblot of MMP10 and GAPDH in AW13516 cells stably overexpressing empty vector or *MMP10*. Numbers on the blot indicate intensity ratio of MMP10 expression with respect to the vector control lane. **b, f, j** Wound-healing assay of AW13516 cells stably overexpressing empty vector or *MMP10* (**b**), AW8507 cells (**f**) and CAL27 cells (**j**) with non-targeting shRNA (sh-NT) or stable *MMP10* knockdown (3 shRNAs – sh1 or sh-2 or sh-3). Representative images of wound-healing assay at 0 h and 21 h are presented along with the bar plots indicating the percentage of wound closure. **c, g, k** Boyden chamber matrigel invasion assay of AW13516 cells stably overexpressing empty vector or *MMP10* (**c**), AW8507 cells (**g**) and CAL27 cells (**k**) with sh-NT or stable *MMP10* knockdown with sh1 or sh-2 or sh-3. Representative images of crystal violet stained Boyden chamber along with the bar plots depict the percent cell invasion (scale bar = 400 μm). **d, h** qRT-PCR and immunoblot of MMP10 and GAPDH in AW8507 (**d**) and CAL27 (**h**) cells with sh-NT or stable *MMP10* knockdown. Numbers on the blot indicate intensity ratio of MMP10 expression with respect to the sh-NT control lane. **e, i** Cell proliferation assay of AW8507 (**e**) and CAL27 cells (**i**) with sh-NT or stable *MMP10* knockdown with sh-1 or sh-2 or sh-3. Scatter plots indicate the number of live cells on the mentioned day. Data are shown as means ± SD. *p*-values are from Student's unpaired *t*-test and denoted as \**p* < 0.05; \*\**p* < 0.01; \*\*\**p* < 0.001; \*\*\*\**p* < 0.0001. Data shown are representative of *n* = 3 independent experiments.

reporter plasmid pGL3-3'UTR-*MMP10* and *miR-944* was cloned in pcDNA-3.1(-) downstream to the CMV promoter. Transient co-transfection of 293FT cells with pc-DNA-*miR-944*, pGL3-3'UTR-*MMP10* reporter led to a significant decrease (*p*-value < 0.01, Unpaired student t-test) in the luciferase activity as compared to vector control, whereas a significant increase (*p*-value < 0.05, unpaired Student's *t*-test) in luciferase activity was observed post-co-transfection with anti-*miR-944* oligo (Fig. 3a). These results establish that the 3'UTR of *MMP10* is a target of *miR-944*.

Next, to understand the role of *miR-944* in metastasis of tongue cancer, *miR-944* was transiently overexpressed in AW8507 cells, which has endogenously low expression of the *miR-944* and high expression of *MMP10* (Supplementary Fig. 6). Overexpression of the miRNA was confirmed at the transcript level (Fig. 3b). Consistent with the results, overexpression of *miR-944* led to a significant downregulation (*p* < 0.01) in the expression of *MMP10* (Fig. 3c). Further, to assess the role of *miR-944* in tongue cancer, in-vitro cell-based assays were performed. A significant reduction in the proliferation rate (*p* < 0.01) of the cells was observed upon *miR-944* overexpression (Fig. 3d). Interestingly, migration and invasion assays suggested a significant decrease in the migratory and invasive phenotype (*p* < 0.001) of AW8507 cells upon *miR-944* overexpression (Fig. 3e, f). Taken together, these results reveal that *miR-944* is a negative regulator *MMP10* and suppresses the metastatic phenotype of tongue cancer cells upon its upregulation.

Consistent with the miRNA microarray study and biochemical findings, validation of *miR-944* expression in 63 tongue tumor samples suggested significant upregulation (*p* < 0.05) of *miR-944*

in patients with low expression of *MMP10* (based on the median expression) as compared to the patients with high expression of *MMP10* (Fig. 3g), suggesting downregulation of *MMP10* by *miR-944*. Furthermore, qRT-PCR based expression of *miR-944* across 93 tongue tumor patient samples suggests that *miR-944* is significantly downregulated (*p*-value < 0.05) in patients with nodal metastasis (Fig. 3h). Overall, we observed significant downregulation of *miR-944* and upregulation of *MMP10* in tongue cancer patients with nodal metastasis, suggesting that *miR-944/MMP10* is potentially involved in regulating lymph node metastasis.

***MMP10* driven invasion and migration of tongue cancer cells is mediated by the *AXL* signaling pathway.** Transcriptome sequencing was performed to investigate *MMP10* downstream targets involved in the regulation of tongue cancer metastasis. Differential expression analysis using cuffdiff and NOISeq tools revealed 110 statistically significant overlapping differentially expressed genes (−1.5 < log2FC > 1.5; *p*-value < 0.05 or prob > 0.95) in AW13516-*MMP10* overexpression clones compared to vector control cells (Supplementary Table 4). Of the 110 differentially expressed genes, 52 showed upregulation, and 58 showed down-regulation (Supplementary Fig. 7a). Analysis revealed significant upregulation (log2FC > 1.5; *p*-value < 0.05) of metastasis pathway-related genes (*AXL, CDH11, COL1A1, FGFBP1, IL6, IL7R, IL11,* and *MMP2*) upon ectopic overexpression of *MMP10* (Supplementary Fig. 7b and Supplementary Tables 4, 5). Along with metastasis pathway related genes, overexpression of *MMP10* also

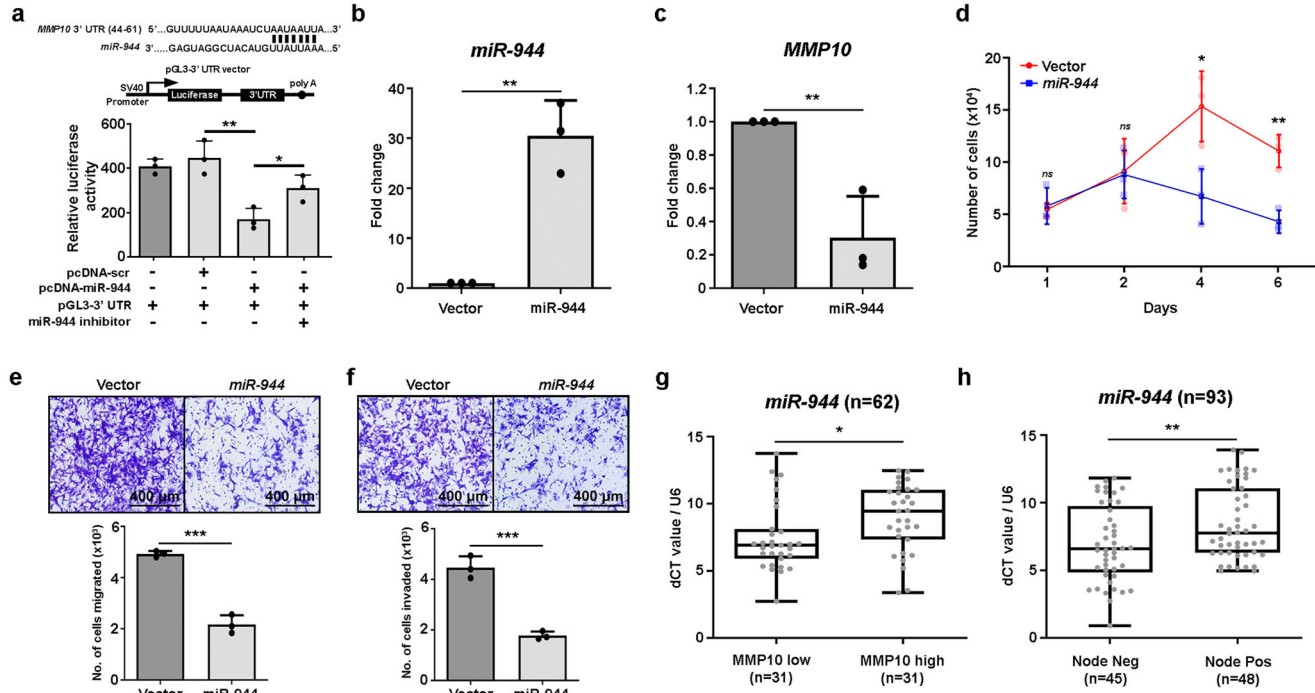

**Fig. 3 *miR-944* targets *MMP10* to suppress metastasis in vitro and in tongue cancer patients. a** Schematic diagram shows the *miR-944* target site on the 3'-UTR region of *MMP10* as predicted by TargetScan. Bar plot represents the relative luciferase activity measured by normalizing with the Renilla luciferase. **b**, **c** qRT-PCR of *miR-944* (**b**) and *MMP10* (**c**) expression in AW8507 cells overexpressing empty vector or *miR-944*. **d** Cell proliferation assay of AW8507 cells overexpressing empty vector or *miR-944*. Scatter plot indicate the number of live cells on the mentioned day. **e**, **f** Cell migration (**e**) and invasion assay (**f**) of AW8507 cells overexpressing empty vector or *miR-944* (scale bar = 400 μm). **g** qRT-PCR of *miR-944* mRNA in 62 primary tongue tumor samples correlated with *MMP10* mRNA expression. **h** qRT-PCR *miR-944* mRNA in 93 primary tongue tumor samples correlated with nodal metastasis status of patients. Data are plotted as boxplot representation of dCT values. *MMP10* and *miR-944* expression are normalized with *GAPDH* and *U6*, respectively. The middle line in the boxplot shows median along with the lower (Q1) and upper quartiles (Q3) as boxes. The whiskers represent the minimum and maximum values. Data are shown as means ± SD. *p*-values are from Student's unpaired *t*-test and denoted as *ns* (not significant); *\*p* < 0.05; *\*\*p* < 0.01; *\*\*\*p* < 0.001. Data shown are representative of *n* = 3 independent experiments.

upregulated EMT markers (*CDH2*, *SNAI2*, Vimentin, MMP9 and β-catenin), and downregulated MET marker (E-cadherin) which were validated in AW13516-MMP10 overexpression clones using real-time PCR or immunoblotting (Supplementary Fig. 7c, d).

Next, we screened for the expression of metastasis pathway-related genes in 52 primary tongue tumors and correlated with *MMP10* expression. With the genes that significantly correlated (*p* < 0.05) with *MMP10* expression (*AXL*, *CDH11*, *FGFBP1*, *IL7R*, and *IL11*) (Supplementary Fig. 8), we performed Kaplan–Meier survival analysis using TCGA-HNSC data. Survival analysis suggests high expression of *MMP10* or *AXL* was significantly (*p* < 0.05) associated with poor overall survival of head and neck squamous cell carcinoma (HNSCC) patients (Supplementary Fig. 9), but not just tongue cancer patients, which could be because there were fewer tongue cancer patients. Also, multiple studies have reported that overexpression of *AXL* promotes metastasis in various cancers, including oral cancer[20,21]. We set to study the role of *AXL* in *MMP10*-promoted invasion and migration of tongue cancer. We screened for the expression of AXL transcript and protein in tongue cancer cell lines (Supplementary Fig. 10a, b). We asked if the knockdown of *MMP10* affects *AXL* expression in the cells ectopically over-expressing *MMP10*. We performed siRNA-mediated knockdown of *MMP10* in AW13516-MMP10 overexpression clones and observed *AXL* to be significantly downregulated (Supplementary Fig. 11a, b), confirming that *MMP10* regulates *AXL* expression. Further, we screened for the expression of AXL downstream signaling molecules (AKT, mTOR and NF-κB p65) upon overexpression and knockdown of *MMP10*. Results suggest that

overexpression of *MMP10* upregulates *AXL* and activates downstream signaling pathway whereas stable knockdown of *MMP10* suppresses these effects (Fig. 4a). Furthermore, we investigated whether *AXL* overexpression in *MMP10* knockdown clones or *AXL* knockdown in *MMP10* overexpression clones would reverse the activation or inactivation of AXL downstream signaling, respectively. We performed stable overexpression of *AXL* in AW13516 (cell line with endogenously low levels of *MMP10*) and AW8507-MMP10 knockdown cells, and siRNA-mediated knockdown of *AXL* in AW13516-MMP10 overexpression clones. We found that overexpressing *AXL* activates AXL signaling in *MMP10* knockdown cells whereas knocking down *AXL* inactivates AXL signaling in *MMP10* overexpressing cells (Fig. 4b), indicating that AXL signaling is downstream of *MMP10*.

To study if *AXL* regulates *MMP10*-mediated proliferation, migration, and invasion of cells in the downstream, *AXL* overexpression and knockdown clones were used for in vitro metastasis-related cell-based assays. *AXL* knockdown significantly reduced the biochemical levels of p-AKT, p-mTOR and p-NF-κB p65 levels and impeded the *MMP10*-driven proliferation, migration, and invasion phenotypes, whereas *AXL* overexpression clones restored the p-AKT and p-NF-κB p65 levels along with proliferation, migration, and invasion phenotypes of *MMP10*-knocked down AW8507 cells (Fig. 4b–k), indicating that *AXL* is downstream of *MMP10* and could potentially regulate the ability of tongue cancer cell lines to metastases upon *MMP10* upregulation. Taken together, these findings indicate that *MMP10*-mediated migration and invasion of tongue cancer cells

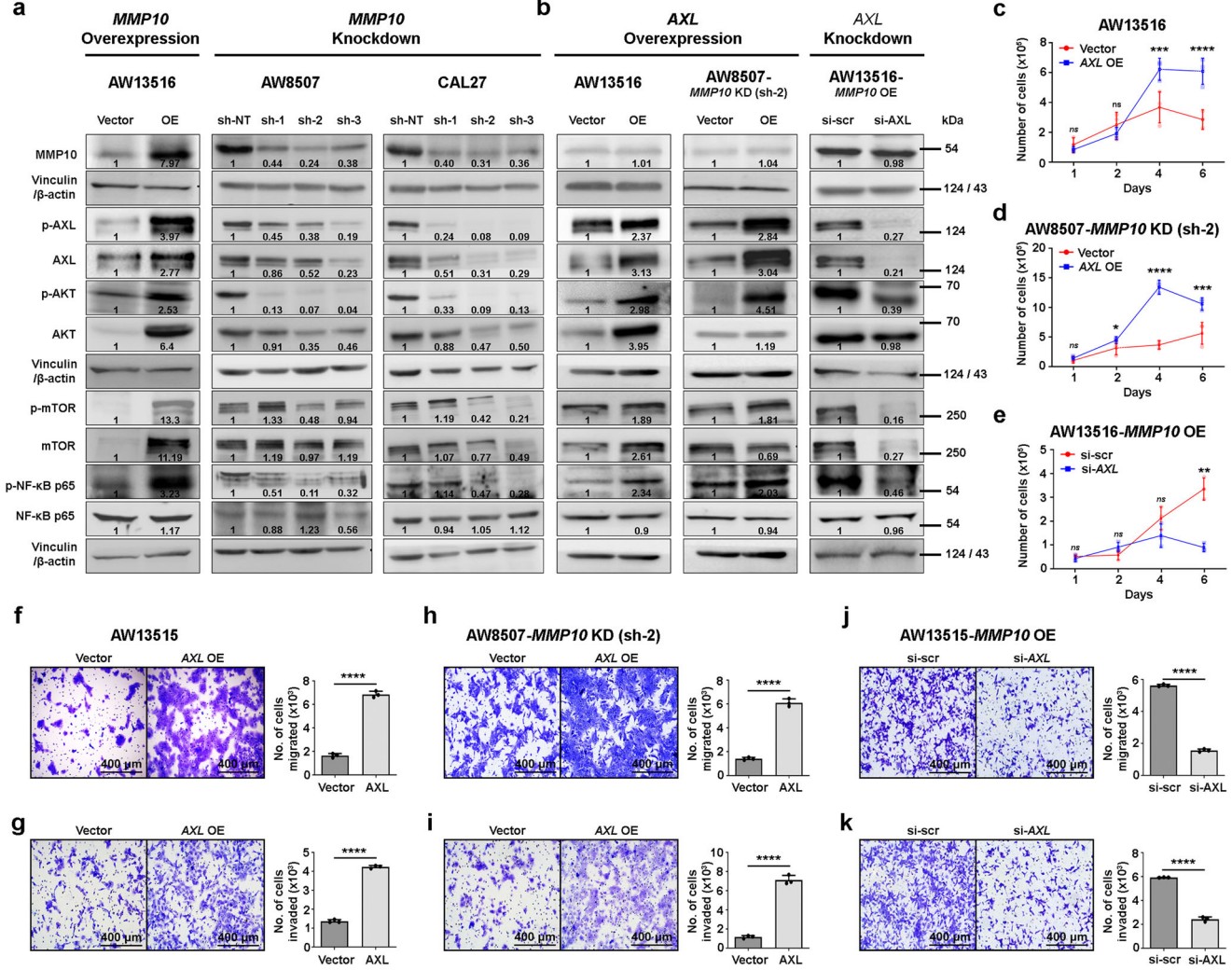

**Fig. 4 miRNA-944/MMP10 induced EMT is mediated by the AXL signaling pathway. a, b** Immunoblot of MMP10, p-AXL, AXL, p-AKT, AKT, p-mTOR, mTOR, p-NF-κB p65, NF-κB p65, Vinculin and β-actin upon overexpression/ knockdown of *MMP10* (**a**) or overexpression/ knockdown of *AXL* (**b**) in tongue cancer cell lines. Vinculin or β-actin was used as loading control. Numbers on the blot indicate intensity ratio of target protein expression with respect to the vector control lane. **c–e** Cell proliferation assay of AW13516 cells stably overexpressing empty vector or *AXL* (**c**), AW8507-*MMP10* knockdown cells stably overexpressing empty vector or *AXL* (**d**) and AW13516-*MMP10* overexpression cells with scrambled siRNA (si-scr) or siRNA-mediated knockdown of *AXL* (**e**). Scatter plots indicate the total number of live cells on the mentioned day. Boyden chamber migration (**f, h, j**) and invasion assays (**g, i, k**) of AW13516 cells stably overexpressing empty vector or *AXL* (**f, g**), AW8507-*MMP10* knockdown cells stably overexpressing empty vector or *AXL* (**h, i**) and AW13516-*MMP10* overexpression cells with scrambled siRNA or siRNA-mediated knockdown of *AXL* (**j, k**) plotted as bar plots indicating the percentage of cells migrated or invaded, respectively. Representative images of crystal violet stained Boyden chambers are shown (scale bar = 400 μm). Data are shown as means ± SD. *p*-values are from Student's unpaired *t*-test and denoted as *ns* (not significant); *$p < 0.05$; **$p < 0.01$; ***$p < 0.001$; ****$p < 0.0001$. Data shown are representative of $n = 3$ independent experiments.

is regulated by activation of the AXL signaling pathway and upregulation of EMT marker genes.

**The orthotopic tongue tumor mouse model validates the role of *MMP10* in nodal metastasis.** To confirm the in vitro findings and study the role of *MMP10* in tumorigenesis and metastasis of tongue cancer in vivo, we established orthotopic tongue tumor mouse model using the *MMP10* overexpression and knockdown clones. We injected luciferase-tagged AW13516 clones of *MMP10* overexpression, and CAL27 and AW8507 clones of *MMP10* knockdown (sh-2) along with vector control cells orthotopically into the tongue of 6 mice/group using needle gauge (30 G) after anesthetizing the mice with isoflurane. Caliper measurements at regular intervals showed that AW13516 cells form tumors after

25 days, CAL27 cells form tumors after 10 days, whereas, AW8507 cells formed tumors after 35 days of injection. However, the volume of AW8507-induced tumors was higher than CAL27-induced tumors, followed by AW13516-induced tumors. Interestingly, knockdown of *MMP10* significantly reduced the tumor volume in both AW8507-induced and CAL27-induced compared to the control group (Supplementary Figs. 12a, b), suggesting the role of *MMP10* in tumorigenesis of tongue cancer. Also, AW13516- and AW8507-induced tumors metastasized to cervical lymph nodes, lung, liver and kidney whereas, CAL27-induced tumors did not metastasize to any organ or lymph nodes (Supplementary Figs. 12c, 13a–c). Of note, the CAL27 cell line is reported to be a non-metastatic cell line in literature as well[22]. In AW13516-induced tumors, metastasis was observed in 6/6 (100%) mice in the *MMP10* overexpression group and 1/6

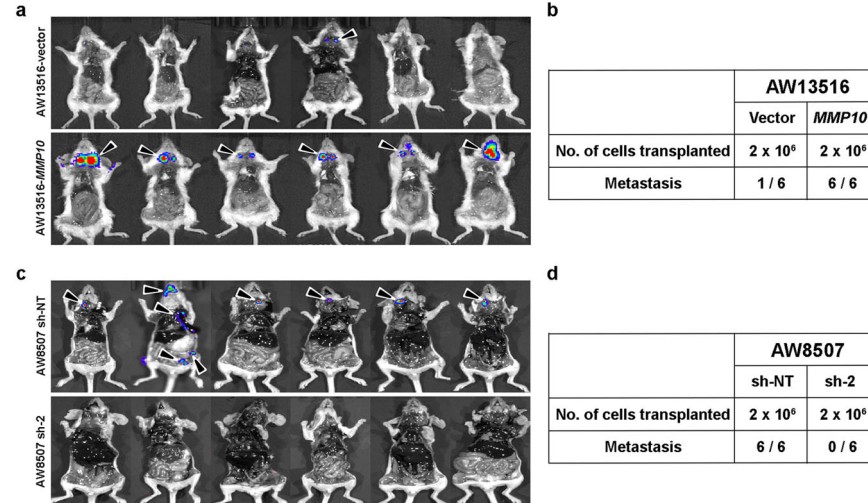

**Fig. 5 MMP10 promotes metastases of tongue cancer in vivo. a, c** IVIS imaging for detection of metastasis in mice injected (n = 6/group) with AW13516 cells stably overexpressing empty vector or *MMP10* (**a**) and AW8507 cells with sh-NT or *MMP10* knockdown (**c**). Bioluminescence imaging was performed after the resection of primary tongue tumors from the mice. The black arrowhead indicates the metastasis in regional lymph nodes and distant organs. **b, d** Tabular representation of the number of cells injected orthotopically into the tongue of mice and number of mice with metastasis in regional lymph nodes or distant organs.

(16.7%) in vector control group. In AW8507-induced tumors, metastasis was observed in 6/6 (100%) mice in the control group and none (0/6) in the *MMP10* knockdown group (Fig. 5a–d), suggesting a significant increase in the metastasis of tongue cancer upon overexpression of *MMP10* and decrease in metastasis upon depletion of *MMP10*. Overall, we demonstrate with the orthotopic tongue tumor mouse model that *MMP10* regulates tumorigenesis and metastasis of tongue cancer.

## Discussion

The rate of occult lymph nodal metastasis of tongue cancer is between 20–30%[4]. There is an unmet need to understand the underlying mechanism to design an effective therapeutic strategy. Here, we present the molecular mechanism of *miR-944/MMP10/AXL*- axis mediated tongue cancer metastasis. Several lines of evidence suggest the pathway is essential and sufficient to induce nodal metastasis in tongue cancer patients.

Firstly, our immunohistochemical and real-time PCR-based analysis of MMP10 in 208 tongue tumor samples (including 98 N-zero clinical trial samples and 110 fresh-frozen samples) suggests a significant overexpression of MMP10 in primary tongue tumors of node-positive patients ($p < 0.00001$). The results show accurate prediction of nodal metastases in 86% (68 of 79) of node-positive patients based on MMP10 expression with 14% (11 of 79) false negative. While most patients with early stage tongue cancer receive elective neck dissection as standard therapy in India and other counties, testing for MMP10 expression may allow for the substantial majority of pathological node-negative individuals to avoid the invasive surgical operation associated with significant morbidity.

Secondly, ectopic overexpression or shRNA-mediated knockdown of *MMP10* affects cell proliferation, migration, and invasion of tongue cancer cells. These findings are consistent with previous reports where upregulation of *MMP10* has been shown to promote invasion and migration in several cancer types[12,14,23], including HNSCC[13]. Furthermore, as *MMP10* is reported to be upregulated in primary tumors of tongue cancer[11], we investigated for miRNAs predicted to be targeting *MMP10* and regulating its expression in tongue cancer. *In-silico* prediction of the

miRNA binding sites in 3'-UTR of *MMP10* revealed seven miR-NAs potentially targeting *MMP10* and qRT-PCR based analysis of these miRNAs across 20 paired normal primary tongue tumors indicated *miR-944* as potential candidate miRNA showing highest negative correlation with *MMP10* transcript expression. Our study confirms *MMP10* as a direct target of *miR-944* by performing luciferase reporter assay in presence of *miR-944* and *miR-944* inhibitors. Moreover, we show that overexpression of *miR-944* downregulates *MMP10*, thereby, suppressing proliferation, migration, and invasion of tongue cancer cells. *miR-944* is reported to have both tumor suppressor[24] and oncogenic[25,26] potential in various cancer types. Our findings are consistent with previous reports where overexpression of *miR-944* has been shown to suppress the invasion and migration of cells[24,27].

Thirdly, in order to understand the downstream pathway mediating *MMP10*-promoted metastasis, we performed whole transcriptome sequencing of tongue cancer lines overexpressing *MMP10* followed by qRT-PCR based validation across 52 primary tongue tumor samples and identified upregulation of *AXL*, a receptor tyrosine kinase gene. The overexpression of *MMP10* also activated AXL downstream signaling molecules (AKT, mTOR, and NF-κB p65) and upregulated key EMT marker genes (*CDH2*, *SNAI2*, Vimentin, MMP9, and β-catenin) whereas, knockdown of *MMP10* suppressed AXL downstream signaling. Interestingly, knockdown of *AXL* suppresses *MMP10*-promoted proliferation, invasion, and migration of tongue cancer cells. And, overexpression of *AXL* promotes *MMP10*-suppressed proliferation, invasion, and migration of tongue cancer cells. These results are consistent with literature that suggests that overexpression of *AXL* promotes proliferation, migration and invasion whereas, genetic knockdown or pharmacological inhibition suppresses these phenotypes in various cancers[20,28,29], including HNSCC and oral cancer[30,31]. Thus, our findings indicate that *AXL* could be a potential therapeutic target in tongue cancer patients.

Fourthly, TCGA-HNSC data suggests poor overall survival of patients ($p < 0.05$) upon overexpression of *MMP10* or *AXL* expression positively correlates with expression of *MMP10* and poor overall survival of the patients, as shown in various cancer types[32,33], including oral cancer[31]. Interestingly, *miR-944* expression is significantly negatively correlated with *MMP10* expression

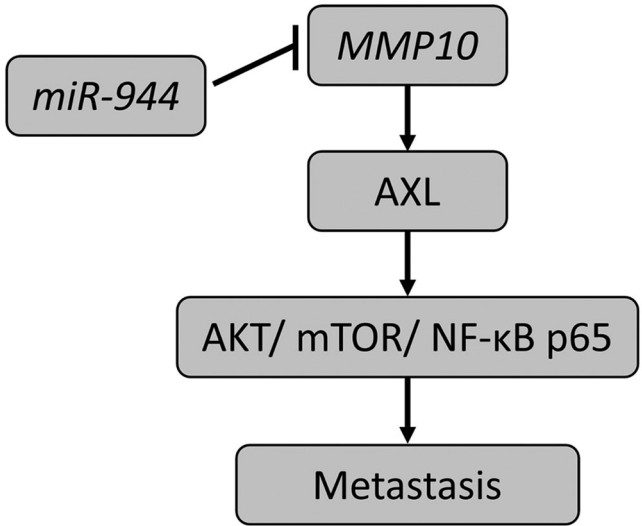

**Fig. 6 Model depicting *miR-944/MMP10/AXL*- axis to regulate metastasis in tongue cancer.** The working model demonstrating the role of *MMP10* in promoting metastasis via activation of AXL signaling pathway and is negatively regulated by *miR-944* in tongue cancer.

($p < 0.05$) and lymph node metastasis ($p < 0.01$), suggesting a significant role of the *miR-944/MMP10/AXL*- axis.

Lastly, we validate our in vitro findings establishing the role of *MMP10* in tongue cancer metastasis using an orthotopic tongue tumor mouse model. We present that overexpression of *MMP10* promotes metastasis (16.7% in WT cells versus 100% in *MMP10* overexpressing cells) and suppression of *MMP10* reduces tumor volume as well as metastasis (100% in WT cells versus 0% in *MMP10* knocked down cells) of tongue cancer in orthotopic tongue tumor mouse model. Although *MMP10* is primarily known for promoting metastatic potential of the tumor cells[13,33], literature also suggests a role of *MMP10* in tumor initiation and progression[12,14]. In cervical and ovarian cancer, *MMP10* is known to be involved in tumorigenesis and metastasis[12,14].

Taken together, our findings describe the role of *miR-944/ MMP10/AXL*- axis in lymph node metastasis in tongue cancer (model summarizing the axis is depicted in Fig. 6). Targeting *AXL* may represent a novel therapeutic approach in managing tongue cancer patients. Additionally, previous studies to stratify the patients likely to develop nodal metastases–not detectable by imaging methods or physical examination, have documented diagnostic limitations. Its reduction could eliminate misdiagnosis sparing 70 to 80% of the patients from morbid elective neck dissection. Thus, low expression of *miR-944* or overexpression of MMP10 as a prognostic biomarker could also help predict nodal metastasis in tongue cancer, an assessment that awaits designing a larger prospective randomized clinical trial.

## Methods

**Patient sample details**. A total of 130 fresh frozen tongue tumor samples were used to screen for the expression of *MMP10* and *miR-944* by quantitative real-time PCR and an additional set of all available 98 formalin-fixed paraffin-embedded (FFPE) tissue blocks with adequate tumor content and quality, as a part of the N-zero clinical trial, were used for screening of MMP10 protein expression by immunohistochemistry. Primary tongue tumors were staged as T1 (measuring ≤2 cm) or T2 (measuring >2 cm but <4 cm) as per AJCC (American Joint Committee on Cancer)/UICC (Union for International Cancer Control) TNM classification system (8th edition). Patients were classified as N0 (node-negative) or N1 (node-positive) status based on the final histopathology report of the primary tumor. Patient sample details are provided in Table 2. The patient details include age, gender, anatomic site, TNM tumor stage, nodal status, smoking (represents the patients with habits of smoking tobacco as cigarettes, cigars or pipes), chewing, alcohol, tobacco (represents the patients with the habit of smoke-less tobacco chewing), recurrence, metastasis and status at last follow-up (the last follow-up date of patients ranges from 1 month to 11.9 years with a median follow-up of 3.8 years. As the patient samples were collected retrospectively, the follow-up period was not reached to 5-year mark in all patients).

**Ethical approval**. All the patient samples were collected from the tumor tissue repository of Tata Memorial Hospital (TMH-TTR) and the Advanced Centre for Treatment, Research and Education in Cancer (ACTREC-TTR), Mumbai. Samples were collected with the approval of The Institutional Review Board (IRB) and the Ethics Committee (EC) of Tata Memorial Centre-ACTREC. Since the samples were collected retrospectively, the IRB and EC waived the need for informed consent.

**Cell culture**. AW13516 and AW8507 tongue cancer cell lines were obtained from Tata Memorial Hospital while CAL27 cells were procured from ATCC. Cells were tested for mycoplasma and were made mycoplasma-free using the EZKill mycoplasma removal reagent (Cat No. CCK006-1, HiMedia). The cell lines were authenticated by DNA short tandem repeat (STR) profiling using Promega Geneprint 10 system in conjugation with GeneMarker HID software tool. All the cell lines (AW13516, AW8507 and CAL27) were cultured in Dulbecco's Modified Eagle Medium (Cat No. 12800-017, Gibco) at 37 °C in a 5% $CO_2$ incubator. Culture media was supplemented with 10% fetal bovine serum (FBS) (Cat No. 10270106, Gibco) and 1.25 µL/mL gentamycin (Abbott). Trypsinization was performed using 0.25% Trypsin-EDTA (Cat No. TC245, HiMedia) and cells were suspended in 90% FBS and 10% DMSO (Cat No. D5879, Sigma-Aldrich) freezing mix for long term storage in liquid Nitrogen.

**Tissue processing, RNA extraction, and real-time PCR**. RNA extraction from the tissue samples was performed using AllPrep DNA/RNA/miRNA Universal Kit (Cat No. 80224, QIAGEN). About 20–30 mg tissue sections were cut into small pieces and subjected to bead-based homogenization in 600 µL lysis buffer using

| Table 2 Clinical characteristics of 228 patients in the study. | | |
|---|---|---|
| **Clinicopathological features** | **Variable** | **Frequency ($n = 228$), n (%)** |
| Age | Median (range) | 46 (23–76) |
| | <40 | 61 (27%) |
| | 40–60 | 125 (55%) |
| | >60 | 32 (14%) |
| | NA | 10 (4%) |
| Gender | Male | 165 (73%) |
| | Female | 53 (23%) |
| | NA | 10 (4%) |
| Anatomic site | Oral Tongue | 228 (100%) |
| TNM Tumor stage | pT1-T2 | 167 (73%) |
| | pT3-T4 | 61 (27%) |
| Nodal Status | Node positive | 83 (36%) |
| | Node negative | 145 (64%) |
| Smoking | Smoker | 53 (23%) |
| | Non-smoker | 165 (73%) |
| | NA | 10 (4%) |
| Chewing | Chewer | 119 (53%) |
| | Non-chewer | 99 (43%) |
| | NA | 10 (4%) |
| Alcohol | Yes | 43 (19%) |
| | No | 175 (77%) |
| | NA | 10 (4%) |
| Tobacco | Yes | 135 (60%) |
| | No | 83 (36%) |
| | NA | 10 (4%) |
| Recurrence | Yes | 116 (51%) |
| | No | 102 (45%) |
| | NA | 10 (4%) |
| Metastasis | Yes | 49 (21%) |
| | No | 169 (75%) |
| | NA | 10 (4%) |
| Status at last follow-up | Alive | 169 (75%) |
| | Died | 49 (21%) |
| | NA | 10 (4%) |

NA = Information not available.

FastPrep homogenizer (MP Biomedicals, USA) and further processed for total RNA extraction as per manufacturer's protocol. Total RNA was extracted from the cell lines using TRIzol reagent (Cat No. T9424, Sigma-Aldrich) based method. Total RNA obtained from tissue samples and cell lines were further subjected to DNase treatment using DNA-free Kit (Cat No. AM1906, Ambion) for removal of the residual genomic DNA. RNA concentration was measured using NanoDrop 2000c Spectrophotometer (ThermoFisher Scientific). First-strand cDNA synthesis was performed using PrimeScript TM 1st strand cDNA synthesis kit (Cat No. RR370A, TaKaRa) as per the manufacturer's protocol. Quantitative real-time PCR was performed using KAPA SYBR real-time PCR master mix (Cat No. KK4601, Sigma-Aldrich) on QuantStudio 12 K Flex Real-Time PCR System (Cat No. 4470935, Applied Biosystems). For quantitative real time PCR of each sample, 6 µL reaction in triplicates were incubated in a 384-well plate at 95 °C for 5 min, followed by 40 cycles of 95 °C for 15 s, 64 °C for 15 s, and 72 °C for 15 s. Primer sequences used for real-time PCR validation of genes are provided in Supplementary Table 6. All real-time PCR experiments were performed in triplicates for each of the three experimental replicates. For real-time PCR analysis, delta-CT (dCt) values were calculated, and samples were sorted into quartiles based on delta-CT values. Samples in the lower quartile were considered as the samples with low expression of MMP10 and the top three quartiles of the samples were considered as the samples with high expression of MMP10.

**Protein sample preparation and western blotting**. Cells were grown to 70–80% confluence in a 10 cm culture dish and washed thoroughly with sterile 1X PBS. The cells were harvested using a sterile cell scraper, and cell lysates were prepared in RIPA Buffer (Cat No. R0278, Sigma-Aldrich) along with a protease inhibitor cocktail (Cat No. P8340, Sigma-Aldrich) and 0.1 M DTT. After intermittently tapping and vortexing the samples on ice for 30 min, cell debris was pelleted by centrifugation at 14000 rpm for 40 min and the supernatant was collected. Protein concentrations were estimated using BCA reagent (Cat No. MP152414, MP Biomedicals). Bovine serum albumin (Cat No. 0216006980, MP Biomedicals) was used as a standard and estimations for each sample was performed in triplicate. For Western blotting, 50 µg of protein was loaded on 10% SDS-PAGE and transferred to PVDF membrane (Cat No. 10600021, Amersham Hybond, GE healthcare) using a wet transfer method. PVDF membrane blocking was performed using 5% BSA to avoid nonspecific binding of antibody. Primary antibodies were diluted in 3% BSA solution prepared in 1X TBST and incubated overnight at 4 °C. Primary antibodies for MMP10 (Cat No. MAB910, Bi Biotech; dilution 1:1000), GAPDH (sc-32233, Santa Cruz Biotechnology; dilution 1:2000), phospho-AXL (DY702, CST; dilution 1:1000), AXL (C89E7, CST; dilution 1:1000), phospho-AKT (4060 T, CST; dilution 1:100), AKT (4685 S, CST; dilution 1:500), Vinculin (4650 S, CST; dilution 1:1000), β-actin (sc-47778, Santa Cruz Biotechnology; dilution 1:2000), phospho-mTOR (5536 T, CST; dilution 1:1000), mTOR (2983 T, CST; dilution 1:1000), phospho-NF-κB p65 (3033 T, CST; dilution 1:2000), NF-κB p65 (ab16502, abcam; dilution 1:2000), Caspase 3 (9662 S, CST; dilution 1:1000), PARP (9542 S, CST; dilution 1:2000), α-Tubulin (T5168, Sigma-Aldrich, 1:2000), E-cadherin (3195 T, CST; dilution 1:2000), Vimentin (5741 T, CST; dilution 1:2000), MMP9 (13667 T, CST; dilution 1:500) and β-catenin (ab32572, abcam; dilution 1:2000) were used for Western blotting. Goat anti-rabbit IgG-HRP secondary antibody (sc-2004, Santa Cruz Biotechnology; dilution 1:2000) was used for the detection of phospho-AXL, AXL, phospho-AKT, AKT, Vinculin, phospho-mTOR, mTOR, phospho-NF-κB p65, NF-κB p65, Caspase 3, PARP, α-Tubulin, E-cadherin, Vimentin, MMP9 and β-catenin. Goat anti-mouse IgG-HRP secondary antibody (sc-2005, Santa Cruz Biotechnology; dilution 1:2000) was used for the detection of MMP10, GAPDH and β-actin. ECL Western Blotting Substrate (Cat No.T7101A, TAKARA) was used for visualization of the luminescence on the Chemidoc system (Bio-Rad). Western blot protein intensities were normalized and quantified using ImageJ software. TIFF files of western blot images were used for quantification. Using the rectangular tool in ImageJ, protein band intensity was measured and normalized with the background intensity in same blot to obtain the normalized band intensity. The actual band intensities were obtained by further normalization with the reference loading control. Fold change expression of each protein was obtained by comparing the actual band intensity of the protein in the overexpression/ knockdown group versus vector control group. All the western blots were performed in three experimental replicates.

**MMP10-miRNA target prediction and Luciferase assay**. The miRNAs targeting MMP10-3' UTR were predicted using the following databases and web servers: MirWalk[34], miRanda[35], mirbridge[36], miRDB[37], miRMap[38], miRNAMap[39], PicTar2[40], PITA[41], RNAhybrid[42] and Targetscan[43]. The primer sequences of miRNAs used for real-time PCR validation are provided in Supplementary Table 6. Real-time PCR data analysis was performed using the $2^{-\Delta\Delta Ct}$ method and U6 was used as internal reference control. For luciferase assay, 293FT cells (50,000 cells/ well) were seeded in a 24 well plate, 12 h prior to transfection. The transfection was performed using lipofectamine 3000 reagent (Life Technologies, USA) with a combination of miR-944 constructs, along with Renilla luciferase vector (for normalizing transfection efficiency) and 15 pmol mirVana miRNA-inhibitor (Ambion, USA) per well. The cells were lysed post 48 h of transfection and luciferase assay was performed to measure luminescence using a luminometer (Berthold Luminometer, Germany). The data were plotted as the ratio of

luminescence of firefly rennila luciferase. The experiment was performed in three replicates, and p-value < 0.05 (calculated using unpaired Student's t-test) was considered significant.

**Cloning of MMP10 cDNA, MMP10-3'UTR and miR-944**. cDNA of human MMP10 was amplified from CAL27 cell line using KAPA Taq DNA polymerase (Cat No. KK1024, Sigma-Aldrich) and cloned in pTZ57R/T cloning vector (InsTAclone PCR cloning kit, Cat No. K1214R, ThermoFisher Scientific) as per manufacturer's protocol. Full-length MMP10 cDNA sequence was confirmed by Sanger sequencing and was sub-cloned into a retroviral expression vector, pBABE-puro (Addgene plasmid # 1764)[44], using restriction digestion based cloning with BamHI-HF (Cat No. R3136, NEB) and EcoRI (Cat No. ER0271, Fermentas) restriction enzyme sites. Similarly, complete MMP10-3'UTR was amplified by designing the primers flanking the 3'UTR with XbaI site for cloning in pGL3-promoter vector (Luciferase Expressing vector, Promega) at XbaI site. Sequencing of the construct was done to ensure the absence of any mutation. For cloning of miR-944, primers were designed to amplify the 432 bp flanking the miR-944 seed sequence from the genomic DNA. The amplicon was sequence verified for mutations using Sanger sequencing and cloned in a pTZ57R/T vector (Fermentas, USA) followed by subcloning in pcDNA 3.1 (-) (ThermoFisher Scientific) using BamHI and HindIII restriction enzyme sites. The primers used for cloning are provided in Supplementary Table 7.

**Transcriptome sequencing and data analysis**. Transcriptome sequencing libraries were constructed by using the TruSeq RNA library protocol (Illumina). Briefly, mRNA was purified from 4 µg of intact total RNA using oligodT beads, and library preparation was done as per the manufacturer's instructions (TruSeq RNA Sample Preparation Kit, Illumina). Transcriptome sequencing was performed and generated at least 43 million reads per sample. Transcriptome sequencing data analysis was performed using the Tuxedo-suite pipeline[45]. In brief, alignment of short reads was done against the reference genome (GRCh38) using TopHat2 in which 95% of reads were mapped to the reference genome. Cufflinks v.2.0.2 was used to find the expressed transcripts in the data and cuffdiff was used for the identification of differentially expressed genes. Also, NOISeq[46] was used for differential expression analysis and overlapping genes were prioritized for functional experiments. Differentially expressed genes were used for the identification of enriched or depleted pathways by Reactome pathway analysis tool[47].

**miRNA microarray analysis**. Microarray-based global miRNA expression profiling of 5 primary tongue tumors derived from patients with no regional lymph node metastases (pT2N0); 4 primary tumors from patients with single lymph node metastasis (pT2N1); and, six adjacent tongue normal samples were analyzed using Affymetrix GeneChip miRNA 3.0 array (Affymetrix, Santa Clara, CA, USA). A 200 ng of total RNA was labeled with biotin using 3DNA Array Detection Flash-Tag™ Biotin HSR kit (Genisphere, Hatfield, PA, U.S.) as per manufacturer's instructions and subsequent hybridization for 16 h at 60RPM. The GeneChip® miRNA 3.0 array contained 25,146 probe sets including 5,639 human miRNAs probe set consisting of mature miRNAs, pre-miRNAs, snoRNAs, scaRNAs, and hypothetical miRNAs. The chip was washed and stained using Gene Chip Hybridization Wash and Stain Kit (Affymetrix) and was then scanned with the Affymetrix GeneChip Scanner 3000 7 G (Affymetrix, Santa Clara, CA, U.S.). All raw microarray data (.cel files) were pre-processed using RMA method[48] which includes background correction, log transformation, quantile normalization followed by essential quality control steps. Probe-centric data were converted into miRNA-centric data using an average over-replicate approach. We excluded snoRNA, hypothetical miRNA, control probes from the analysis and restricted analysis only for annotated microRNAs (miRBase-20 release), and the data was analyzed using BRB-ArrayTools developed by Dr. Richard Simon and the BRB-ArrayTools Development Team. Briefly, non-variable miRNAs were excluded from the analysis based on the log expression variation filter (variance of miRNAs across the arrays). The microRNAs were considered as differentially regulated if miRNA followed 0.5 < fold change > 1.5 filters along with p-value < 0.05. Class comparison analysis was carried out between tumors with node-positive status versus tumors with node-negative status. Class comparison results were plotted as a heatmap in MultiExperimentViewer (MeV version 4.9).

**siRNA synthesis by in vitro transcription**. Sense and anti-sense DNA oligonucleotides for AXL and MMP10 (shown in Supplementary Table 8) were ordered from Sigma-Aldrich. The protocol for oligonucleotide-directed production of small RNA transcripts with T7 RNA polymerase is reported in the literature[49]. For each in vitro transcription (IVT) reaction, 1 nmol of each oligonucleotide (re-suspended in 1X TE buffer (10 mM Tris-HCl pH 8.0 and 1 mM EDTA)) was annealed using thermocycler to obtain double-stranded DNA (dsDNA). The thermocycler conditions used were: 95 °C for 3 min, followed by 70 cycles of 95 °C for 30 s (−1 °C/ cycle). In vitro transcription (IVT) reaction was performed in 20 µL of a reaction containing 1X T7 transcription buffer (Cat No. P118B, Promega), 1X biotin RNA labeling mix (Cat No. 11685597910, Sigma-Aldrich), 1 U RiboLock RNase Inhibitor (Cat No. EO0381, Fermentas), 10 U T7 RNA polymerase (Cat No. P2075, Promega) and 1 nmol of dsDNA, as a template. The reaction was incubated at

37 °C for 2 h. Sense and anti-sense siRNAs synthesized in separate reactions were annealed by mixing the transcription reactions at 95 °C for 1 min, followed by 70 cycles of 95 °C for 30 s (−1 °C/cycle) to obtain double-stranded small interfering RNA (siRNA).

**Overexpression and knockdown studies**. For overexpression of *MMP10* in the AW13516 cell line, the pBABEpuro-*MMP10* construct was used. Cells with pBABE-puro empty vector were used as a control for overexpression. For knockdown of *MMP10* in CAL27 and AW8507 cell lines, 3 lentiviral shRNA constructs (Transomic Technologies) were used. The short hairpin non-targeting (sh-NT) construct was used as vector control. Transfection was carried out using a lipofectamine kit (Cat No. L3000015, Invitrogen). Positive clones were selected using 0.5 µg/mL of puromycin (Cat No.TC198, Himedia). For stable verification of Myristoylated *AXL* in AW13516 and AW8507-*MMP10* knockdown (sh-2) cells, pWZL-Neo-Myr-Flag-*AXL* construct[50] was used. The construct was obtained as a kind gift from Dr. Shaida Andrabi (University of Kashmir, Jammu and Kashmir, India). Positive clones were selected using 1 mg/mL of G418 (Cat No. TC025, HiMedia). Transient siRNA-mediated knockdown of *MMP10* or *AXL* was performed with two siRNAs, targeting each gene, synthesized using T7 RNA polymerase. Following siRNA transfection for 48 h using Lipofectamine RNAiMAX (Cat No. 13778075, Invitrogen), cells were used for RNA isolation and in vitro cell-based assays. Transient overexpression of *miR-944*-pcDNA 3.1(-) was carried out in AW8507 cells. Transient transfection was performed by using lipofectamine (Cat No. L3000015, Invitrogen) and cells were collected after 48 h for RNA isolation and to perform cell-based assays. All the transient knockdown and overexpression experiments were performed in three independent replicates and the expression of target gene was screened before performing the cell based assays.

**Cell Proliferation assay**. Cell proliferation assay was performed in 24 well plates with a cell density of 10,000 cells/well for all the experiments. Cell growth was assessed at days 1, 2, 4, and 6 by passaging and counting viable cells by trypan blue staining and using a hemocytometer. All experiments were performed three times with each experiment performed in triplicates.

**Wound-healing assay**. An equal number of cells were seeded in 6-well tissue culture plate (3 wells per each clone) for control and overexpression or knockdown clones and cultured till the plate was confluent. Confluent monolayers in the plate were subjected to scratch (3 scratches for well) with a sterile pipette tip. Further, cells were washed with 1X PBS to remove debris and subsequently incubated with DMEM medium containing 10% FBS. Cell migration at the wound surface (3 wound per well with 3 wells per each clone = 9 wounds for each clone in one experiment) was imaged for a period of 21 h under an inverted microscope with one image at every 30 min. The quantification of cell migration was performed using the ImageJ wound healing plugin tool by measuring the distance of the wound edge of the migrating cells from the start point to the migrated point in nine separate wounds per each clone in one experiment. All the experiments were independently performed three times.

**Boyden chamber invasion and migration assay**. Boyden chamber matrigel invasion assay was performed using 24-well transwell inserts (Cat No. 353097, Corning) coated with 100 µg matrigel (Cat No. 354234, Corning) and allowed to settle for 16 h at 37 °C in 5% CO$_2$ incubator. Invasion assay was performed with $1 \times 10^5$ cells of AW8507-*MMP10* knockdown clones, $5 \times 10^5$ cells of CAL27-*MMP10* knockdown clones, $2 \times 10^5$ cells of *AXL* overexpression or knockdown clones and $4 \times 10^5$ cells of *miR-944* overexpression clones suspended in 300 µL internal serum-free medium and seeded in the Boyden chamber and 700 µL of 10% serum-containing DMEM medium was added in the companion plate wells. For migration assay, cells were seeded directly in the Boyden chamber without matrigel. For migration assay, $1 \times 10^5$ cells of *AXL* overexpression or knockdown clones and $2 \times 10^5$ cells of *miR-944* overexpression clones were used. The cells were allowed to invade/migrate for 48 h at 37 °C in 5% CO$_2$ incubator. The transwell chambers were fixed and stained with 0.1% crystal violet. After mounting the membrane using DPX mountant (Cat No. 18404, Qualigens) on a slide, invaded cells were imaged under an upright microscope at 10X magnification. Images of 10 random fields were chosen and the number of cells in each field were counted using the ImageJ cell counter plugin tool and plotted as percent cell invasion or percent cell migration. All the experiments were independently repeated three times with 2 inserts/clone in each experiment (total of 6 replicates per each clone).

**Apoptosis assay**. Apoptosis was measured by double-labeling the cells with Annexin V-fluorescein isothiocyanate (FITC) and propidium iodide using FITC Annexin V Apoptosis Detection Kit, as per the manufacturer's instructions (Cat No. 556570, BD Pharmingen). Briefly, 0.3 million cells of AW8507- and CAL27-*MMP10* knockdown along with scrambled control cells were re-suspended in 100 µL 1X Annexin V binding buffer and incubated on ice with 2 µL Annexin V-FITC conjugate for 15 min. This suspension was diluted to a final volume of 300 µL using ice-cold 1X Annexin V binding buffer and acquired immediately on Attune NxT Acoustic Focusing Cytometer (Life Technologies, ThermoFisher Scientific) after addition of 1 µL of 100 µg/mL Propidium iodide (PI). Data analysis

was performed using FlowJo software (v10.6.1). All the experiments including siRNA-mediated knockdown of *MMP10* in both cell lines were performed three times.

**Cell viability assay**. Cell viability was assessed by resuspension of 0.3 million cells in 300 µL 1X Annexin V binding buffer and addition of 1 µL of 100 µg/mL Propidium iodide (PI). Percent viable and late apoptotic/necrotic cells were acquired on Attune NxT Acoustic Focusing Cytometer. Data analysis was performed using FlowJo software. All the experiments were performed in three replicates.

**Immunohistochemistry**. Samples used for IHC were duly verified by two independent reviewers for histological examinations such as normal sample verification and percent tumor nuclei. The tumor sample with concordant histopathological diagnosis by both reviewers was included in the study. Immunohistochemistry was performed using the standard protocol of the Vectastain Elite ABC Universal kit (Cat No. PK-6200, Vectorlabs), as also previously reported[51]. Briefly, antigen retrieval was performed by incubating the slides in preheated citrate buffer (pH 6) in a pressure cooker for 10 min. The slides were allowed to cool at room temperature and rinsed with 1X TBST (Tris-Buffered saline with 1% Tween 20). The endogenous peroxidase activity in tissue was blocked by incubating the slides in 3% hydrogen peroxide. The slides were blocked by horse serum for 1 h before incubating with the MMP10 primary antibody (Human MMP10 monoclonal antibody, Cat No. MAB910, Bi Biotech) overnight at 4 °C in a moist chamber. Post incubation, the slides were rinsed with 1X TBST and incubated with a universal secondary antibody (Vectastain Elite ABC Universal kit). The chromogenic reaction was performed using 3,3'-diaminobenzidine chromogen solution for 5 min which resulted in brown color staining. The slides were rinsed in deionized water and counterstained with hematoxylin. Finally, the slides were dehydrated and mounted with DPX mounting medium and coverslip. Further, the immunohistochemistry staining was evaluated. Based on the IHC staining score for MMP10, +3 (strong staining) and +2 (moderate staining) scores were considered as high expression, and +1 (weak staining) and 0 (no staining) scores were considered as low expression of MMP10.

**Orthotopic tongue tumor mouse model and IVIS imaging**. All in vivo experiments were performed as approved by Institutional Animal Ethics Committee (IAEC), TMC-ACTREC. Luciferase-expressing stable clones of AW13516 cells stably overexpressing empty vector or *MMP10*, AW8507 and CAL27 (*MMP10* knockdown (sh-2) and vector control clones). CAL27 ($1 \times 10^6$ cells), AW13516 and AW8507 ($2 \times 10^6$ cells) clones were trypsinized and suspended in 40 µL sterile 1X PBS. Cells were injected orthotopically into the tongue of 6–8 week old female nude mice ($n = 6$/group) using needle gauge (30 G) after anaesthetizing the mice with isoflurane. Mice in the control group were injected with vector control cells. Caliper measurements were performed after every 7 days to monitor the volumes of the tumors using following formula: [length (mm) × ( breadth)$^2$ (mm)$^2$]/2. Tumor formation was confirmed by non-invasive bioluminescence imaging (IVIS Spectrum, In Vivo Imaging System, Caliper Life Sciences) at the Molecular Imaging Facility, ACTREC. Mice were anesthetized by continuous 2% isoflurane inhalation and 100 µL of 30 mg/mL of D-luciferin substrate (D-Luciferin Firefly, potassium salt, Cat No. L-8220, Biosynth Carbosynth) was injected intraperitoneally for bioluminescence imaging (BLI). Living Image 4.5 software was used to measure bioluminescence signal peak using auto exposure mode. For the assessment of metastasis, mice were sacrificed and subjected for BLI imaging after excision of complete tongue with the primary tumor. Further, mice were dissected for ex-vivo BLI imaging of internal organs (submaxillary glands and cervical lymph nodes, lungs, liver, kidney, spleen, uterus and ovaries) collected in petri dish and sprayed with D-luciferin substrate to assess distant metastasis using IVIS system.

**Survival analysis**. Survival analysis was performed using Kaplan–Meier plotter online tool[52] in 499 TCGA-HNSC samples. TCGA-HNSC clinical data was directly imported into the Kaplan-Meier plotter server using the Pan-cancer RNA-seq data. *MMP10* expression status (high or low) was assigned to the samples assessed in the survival analysis based on the "Auto select cut-off" option, which selects the best performing cut-off between the upper and lower quartiles.

**Statistics and reproducibility**. Correlation between expression of genes and nodal status/other clinical parameters, in the in-house and TCGA-HNSC data, was performed using R programming (www.r-project.org). Statistical analysis was performed using GraphPad Prism version 8 software (GraphPad Software, La Jolla, CA). Mann–Whitney test was used to calculate the statistical significance of MMP10 expression in node positive and node negative tongue tumor samples. The student's unpaired *t*-test was used to determine the statistical significance between different groups and the *p*-values calculated are denoted as *ns* (not significant); *$p < 0.05$; **$p < 0.01$; ***$p < 0.001$; ****$p < 0.0001$. Reproducibility of the experimental findings were confirmed by performing $n = 3$ independent replicates of each experiment. The findings of all the biological replicates were consistent.

**Reporting summary**. Further information on research design is available in the Nature Research Reporting Summary linked to this article.

## Data availability

The transcriptome sequencing data generated and analysed during the current study is available in the ArrayExpress repository under the accession number: E-MTAB-11185. The source data behind all the graphs in the paper are provided in the Supplementary Data 1 and the uncropped raw western blot images (Supplementary Figs. 14–25) are provided in the Supplementary Information.

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

## Acknowledgements
We thank all members of the Dutt laboratory for critically reviewing the manuscript. We thank Mr. Mahadev Mandavkar for helping in the standardization of the IHC protocol. Mr. Madan Ludbe, Mr. Anand Pawar, Mr. Ajay Chalke and Mr. Krishna Sawant from ACTREC and TMH for helping in the collection of patient samples. A.D. was supported by an Intermediate Fellowship from the Wellcome Trust/DBT India Alliance (IA/I/11/2500278), and annual intramural institutional funding from DAE 1/3(7)/2020/TMC/R&D-II/8823 and DAE 1/3(6)/2020/TMC/R&D-II/3805. B.D. is supported by a senior research fellowship from CSIR. S.D. is supported by a senior research fellowship from ACTREC. The funders had no role in study design, data collection, and analysis, decision to publish, or preparation of the manuscript.

## Author contributions
B.D. and A.D. conceptualized the study and designed the experiments. B.D., A.B., A.P., S.D., P.U., A.R., R.K., S.M. and R.T. performed the experiments. B.D., A.B., A.P., S.D., A.R. and A.D. performed the data analysis. K.S., R.V., M.B., P.G., A.K.D. and S.N. generated reagents and provided clinical samples. B.D., R.K. and A.D. wrote the manuscript. All authors read and approved the final manuscript.

## Competing interests
The authors declare no competing interests.
