## [Peer Review File · Communications Biology]

Reviewers' comments:

Reviewer #1 (Remarks to the Author):

The study by Dharavath et al. shows the prognostic and cancer-promoting role of MMP10 in tongue cancer and provides evidence for the regulation of MMP10 by miR-944. They also show the connection of MMP10 to AXL and its downstream signaling. The authors have performed extensive set of experiments, their findings are interesting, and the data is mostly presented well. The study uses rather large sets of human samples, which are examined by IHC and qRT-PCR. Various cell experimental and other laboratory experiments are performed to study functional role of MMP10, miR-944 and AXL. Moreover, the authors use biocomputing methods and integrate all into nicely compiled paper. However, the work has some issues that need to be fixed or described more carefully before the work can be properly evaluated and is acceptable for publication.

The role of MMP10 in oral cancer has been shown previously, but the current findings by authors strengthen the previous finding. The miR-944/AXL axis related to MMP10 is novel and overall, the study provides further support for the functional role of MMP10 and miRNAs in tongue cancer. Thus, the paper would add interesting knowledge in the field for the function of MMP10 and miRNAs in oral cancer and the mechanisms on how the effects of MMP10 are mediated and regulated.

Experiments are well performed, and in most parts enough information is provided for the methods and about the data acquired. Also, mostly, enough experiments have been performed to support the conclusions, although some claims are too strongly expressed and need to be toned down. Some descriptions of, for example, statistical analyses and experimental setups (repeats, number of samples in each experiment) and the data of western blots need to be improved in order to evaluate the data properly. More specifically:

1. Please tone down the title as there is not enough in vivo evidence for such strong conclusion – although the role of miR-944 and AXL were examined in vitro, their role was not examined in mouse model to conclude them as mediators of LN metastasis.
2. Abstract says: 'Next, using an orthotopic tongue cancer mouse model, we show that loss of MMP10 or miR-944 overexpression suppresses proliferation, migration, and invasion of tongue cancer cells impeding nodal metastasis.' This sentence should be changed as it gives an impression that all this was shown by mouse model. Instead, mouse model only provided data for the role of MMP10 and in vitro evidence (or partly human samples) supported the rest.
3. Page 15. row 346: ...'regulates lymph node metastasis.' please tone down as the conclusion is too strong based on the data.
4. Conclusion that AXL is connected to miR-944 cannot be made solely based on the connection of MMP10 with miR-944 and MMP10 with AXL separately (abstract, results p15. row. 348 title)
5. Page 3. row 70: lymphogenesis \diamond lymphangiogenesis. Please correct.
6. 'N-zero clinical trial', please provide short explanation for this.
7. Page 20, row 453. 'Anti-correlate' maybe other expression should be used?

8. Supplement SFigure3 is mentioned before SFigure2. Please check for the correct order.
9. Did the authors consider other methods to study proliferation as passaging and counting by hemacytometer does not accurately measure proliferation (could be mix of differences in viability, proliferation, apoptosis) and is prone to errors. Did the authors test differences in apoptosis separately?
10. Some methods properly describe 1) the number of individual experiments performed and 2) number of technical repeats in each experiment. However, some lack this information. Please add this information for all experiments. Without this information, it is impossible to evaluate the results.
11. There is some uncertainty (partly because of the above-mentioned shortages on repetitions of experiments) on how statistical analyses were performed in each experiment. For example, in Figure 2, many of the Figures show relative values to control (=1 or 100). From which values were the P values calculated? It would be more informative to illustrate the values used for statistical analyses in Figures and not the ratios to control. Also, for 2E, 2I, the statistics are confusing. Are the controls (values from technical repeats, individual experiments?) compared to all sh cells? The differences illustrated by these experiments seems clear, but statistics should be performed properly and described thoroughly. Without this information, it is impossible to evaluate the results.
12. Do the authors have reason to use P values <0.005 for ** and <0.0005 for *** as more commonly used values would be <0.01 and <0.001.
13. For Western blots, molecular weight standards should be marked (for all) figures and original, whole blot images should be provided as Supplementary data. Without this information, it is impossible to evaluate the results.
14. How many times the blots were performed? For at least most important differences claimed based on the results, three blots should be performed and the differences in band intensities measured and related to housekeeping protein.
15. In Figure 2, two separate blots for vinculin as housekeeping are shown. However, twelve proteins were detected and presumably, they are not all from two different blots with vinculin in addition. If this is the case, housekeeping should be provided for all blots separately. Without this information, it is impossible to evaluate the results.
16. IHC: the authors do not provide information for the antibody used? Should there be also isotype control to show the antibody specificity – that depends on the antibody used. Without this information, it is impossible to evaluate the results.
17. The authors should describe in the text (and for this, please provide higher magnification images along with the current ones) where is the MMP10 IHC staining localized – is it inside the cells, on membrane, extracellular? Also, the distribution of MMP10 in tumors would be highly informative and interesting – e.g. is it expressed in the invasive front, close to certain tissue structures etc.?
18. There are some errors in reference list, e.g. Ramqvist, T. et al. MMP-10/Stromelysin-2 Promotes Invasion of Head and Neck Cancer. PLoS ONE 539 6, e25438, doi:10.1371/journal.pone.0025438 (2011). The first author is incorrect.

Reviewer #2 (Remarks to the Author):

In this study, the authors examined the role of miR-944/MMP10/AXL-axis in node-positive tongue tumors. Authors found that MMP10 overexpression was frequently observed in node-positive tongue

tumors and knockdown of MMP10 suppressed proliferation, migration, and invasion. Moreover, authors showed that miR-944 negatively regulates MMP10 by targeting its 3'-UTR. Among upregulated genes by MMP10 overexpression, authors focused on AXL and miR-944/MMP10/AXL-axis was involved in tumor progression both in vitro and in vivo. In conclusion, authors establish that the miR-944/MMP10/AXL- axis underlies lymph node metastases with potential therapeutic intervention and prediction of nodal metastases in early-stage tongue cancer patients.

It is well-known that MMP10 is involved in tumor progression. In this study, it is still unclear the detailed mechanism of miR-944/MMP10/AXL-axis in node-positive tongue tumors via promoting proliferation, migration, and invasion. I feel that this paper is too preliminary for publication.

My comments are as the following:

1. In Figure 1, authors showed that MMP10 overexpression was involved in tongue cancer patients with lymph node metastasis. However, authors previously reported that MMP10 overexpression was associated with lymph node metastasis in early tongue cancer patients. Even if the tumor size is small (T1 and T2), if patients have metastasis, authors cannot say early stage. Authors should not use the term "early stage". Moreover, authors should explain why you focused on T1 and T2 cancer. Moreover, in your previous study, authors used HPV negative early-stage tongue cancer patients habitual of chewing betel nuts, areca nuts, lime or tobacco. In general, cancer patients habitual of chewing betel nuts, areca nuts, lime or tobacco have buccal cancer. Why authors focused on tongue cancer in previous and current studies?
2. Authors should demonstrate that the involvement of miR-944/MMP10/AXL-axis is specific in T1 and T2 tongue cancer. To prove your hypothesis, authors should explain the following question.
 - 1) How about the role of miR-944/MMP10/AXL-axis in T3 and T4 tumors?
 - 2) How about the involvement of miR-944/MMP10/AXL-axis in early-stage cancer (T1-T3 without metastasis)?
 - 3) How about the role of miR-944/MMP10/AXL-axis in the precancerous lesions (ex. epithelial dysplasia)?
 - 4) How about the role of miR-944/MMP10/AXL-axis in HPV-positive cancers?
3. Authors should examine the MMP10 overexpression and check the upregulation of AXL. In addition, authors should check the phenotype of AXL overexpression.
4. Authors showed that MMP10 overexpression promotes proliferation, migration, and invasion via AXL overexpression. How about the mechanism (how AXL regulates proliferation, migration, and invasion)?

Reviewer #3 (Remarks to the Author):

In the manuscript "miR-944/MMP10/AXL- axis Mediates Lymph Node Metastasis in Early-Stage Tongue Cancer" by Dharavath et al, the authors validate the overexpression of MMP10 in tongue tumor sample and correlated with nodal metastasis. They also performed functional analysis in vivo and demonstrated that MMP10 is essential for tumor growth and nodal metastasis in mouse model. The authors also studied the mechanistic effect of MMP10 upregulation and demonstrated that miR-944 negatively regulates MMP10, and AXL signaling pathway mediates phenotypes associated suppressing proliferation, invasion, and migration of target tongue cancer.

Although the study is comprehensive and the amount of data generated are valuable, some concerns should be clarified in order to improve the quality and the impact of the manuscript.

Major comments:

The main criticism is that some experiments, specially those focusing in the mechanism of MMP10 regulation in tongue cancer, lacks the appropriate control and some data are not consistent through the manuscript.

1-) The main finding in the paper relies on the proposed regulatory mechanism of miR-944/MMP10/Axl axis. In this regard, it is not clear why the authors did not evaluate the basal expression level of Axl in all cell lines employed in this study. For instance, only the levels of MMP10 were demonstrated (Figure S2). However, the authors did not show the levels of Axl in the cell. Does Axl levels correlate with MMP10 levels in all cell lines? And in tissue samples?

2-) What are the activation status (phosphorylated form) of Axl in the cell lines? And in the tissue samples?

3-) The mechanistic analysis was performed in MMP10 overexpressed cells. Considering that AW8507 and CAL27 express high levels of MMP10, why the authors overexpress this protein in a cell line and did not evaluate the mechanism in AW8507 and/or CAL27?

4-) Since the authors are claiming that MMP10 induces Axl expression, it would be important to demonstrate whether MMP10 indeed modulates Axl and proteins in the Axl pathway expression. In this regard, would be interesting to use MMP10 knocked down and evaluate Axl and Axl pathway protein levels

5-) There is lack of consistency in the western blots. MMP10 expression levels are completely different for cell line AW13516 in figure 3I and figure S2C (Levels of MMP10 in figure S2 is different when compared to Figure 3I). Why did the authors used both GAPDH (figure S2C) and vinculin (figure 3I) as loading control?

6-) Quantification and normalization of proteins in the Western blots protein should be performed.

7-) A figure summarizing the regulation of miR-944/MMP10/Axl axis in this model, should be included in the manuscript.

Reviewer #4 (Remarks to the Author):

Review report for Dharavath et al. manuscript (Communications Biology)

This study by Dharavath et al. describes the pro-metastatic functional effects of matrix metalloproteinase 10 (MMP10) on oral tongue cancer (OTSCC) both in vitro and in vivo. Furthermore, they demonstrate the regulation of MMP10 expression by miRNA-944 and discover the downstream signaling molecules (namely AXL), which transmit the functional effects of MMP10. Additionally, the use of MMP10 as a prognostic marker in OTSCC is explored. The researchers have used extensive molecular and cell biology methods as well as bioinformatic tools, which is the strength of this study and increases the quality of the research and reliability of the presented conclusions. The findings in this study are novel related its functional effects of MMP10 on OTSCC cells as well as regulation and signaling of MMP10 in OTSCC. Otherwise, the presented results support the previous findings on this topic. Previous research on this specific topic is limited in general, which also warrants the publication of this study together with its depth and validity. Yet, there are minor corrections to be made (suggestions given below) before a full approval can be given – these corrections should also improve the readability and impact of this manuscript even further.

Abstract:

The abstract is compact and nicely written, but some of the presented conclusions are possibly misleading.

1. Please consider revising the statement on the second sentence (rows 37-38). This result is not exactly clear in results section (see comment x) nor does the given n correspond to the number of node-positive tumors.

2. The sentence starting from row 41 "Next, using orthotopic tongue cancer model..." gives the impression that all functional aspects (proliferation, migration and invasion) were studied in the mouse model, which based on the results, is not the case (they were studied in vitro). Please consider revising this sentence.
3. For Keywords, please consider including AXL or miRNAs instead of "Epithelial mesenchymal transition" as they could better reflect the results and conclusions of this manuscript.

Introduction:

Introduction section gives the necessary overview on the research topic and question by discussing related literature. Yet this section could benefit from few changes as suggested below:

4. LNM is indeed found to result in worse prognosis for OTSCC patients, as suggested by authors in first part of the introduction (rows 60-62). Yet the references used do not discuss OTSCC but other cancers, although such research and associated literature exists. Please consider referring to OTSCC-related publications or modifying the sentence accordingly.
5. Row 70: Please consider the term "lymphogenesis", should it read "lymphangiogenesis" instead?
6. The sentence on rows 74-76 only refers to ovarian cancer, although he referenced studies also include HNSCC and cervical cancer. Please consider revising this sentence.
7. Please consider toning down the conclusion in the sentence on rows 79-81. Even if MMP10 have been upregulated in another head and neck cancer and relates to pre-metastatic niche formation in cancers overall, the variation between cancers overall and within HNSCCs is great. Thus, it is hard to convincingly suggest that MMP10 would play a LNM-related role in OTSCC based on these factors, although it warrants the research on this topic.

Materials & Methods:

In general, the methods are clearly written out. Some clarifications could still be discussed.

8. On patient samples (Table I):

Comment 8.1. This study focuses on tongue cancer, yet the patient cohorts also include tumors from the anatomical site of buccal mucosa. Could the authors explain why and describe, whether these tumors were excluded from the analyses or not?

Comment 8.2. Could the authors describe what is the difference between smoking and tobacco status?

Comment 8.3. What was the last follow-up date of these patients? Did it vary or were all followed to the suggested 5 years mark?

9. For western blotting (page 6 in M&M, page1-2 in supplementary M&M): How was the unspecific antibody binding blocked?

10. In general, the methods describe both technical replicates in assays as well as number of repeated experiments in a varying manner resulting this information to be missing in some assays (e.g. boyden chamber assays). Could the authors please provide this information for all the assays when necessary (this suggestion includes supplementary M&M).

Results:

This study provides a generous number of solid results to support the conclusions drawn in the next chapter.

11. For figure 1B: Could the authors explain why they have presented the gene expression values as negative delta Ct values? This is quite an uncommon presentation.

12. In the sentence on rows 251-254 authors claim OTSCC patients with LNM show higher gene and protein MMP10 expression, but seemingly the data is not provided for RT-PCR analysis. Please check.

13. For Table II: Authors claim significant results in text (rows 264-265), yet the table does not provide these. Please check and if needed, modify this sentence accordingly.

14. For figures 2 and 3: Please provide the molecular weights (or their markers) for the western blot images (this is also a recommendation by the journal). For further validation, whole blot images could also be provided for evaluation.

15. On page 16 authors describe the effect of high MMP10 or AXL expression on HNSC(C) patient survival. Have the authors checked if this conclusion can be made based on only OTSCC patients in this data? HNSCCs vary greatly in etiological and other factors (e.g. HPV), which might also affect

survival.

16. On rows 379-381 authors claim that AXL could regulate metastasis, yet this conclusion is hard to justify only based on results from in vitro assays. Please consider toning down this conclusion.

17. On row 387 authors claim to have developed the orthotopic tongue cancer mouse model when indeed this model has been introduced by Myers JN et al already in their 2002 publication in Clin Cancer Res journal. Please consider using another verb in this sentence.

Conclusion:

18. Could the authors please clarify the comment on "elective neck dissection that is mandatorily performed" (juxtaposition, rows 416-419). Also please note that different countries might have different standards on elective neck dissection which should be taken into account.

19. Please check and possibly revise the use of term anti-correlated (unusual term, row 453) as well as consider if the sentences on rows 460-463 could be merged as they use the same references and discuss the same topic.

References:

Please carefully check and correct the possible errors in references, see at least the following:

20. Reference 3 is apparently a book chapter. Please check the correct formatting for the reference type.

21. Reference 12 (Ramqvist et al. 2011) contains wrong first author.

22. References 37 and 40 are the same reference.

Response to the reviewers:

We thank all the reviewers for their thorough analysis of our work. Please find the specifics of the changes made to the manuscript in blue, in response to the reviewers' recommendations.

Reviewer #1:

Comment #1: Please tone down the title as there is not enough *in vivo* evidence for such strong conclusion – although the role of miR-944 and AXL were examined *in vitro*, their role was not examined in mouse model to conclude them as mediators of LN metastasis.

Response: As suggested by the reviewer, the title of the manuscript is changed to “*Role of miR-944/MMP10/AXL- axis in Lymph Node Metastasis in Early-Stage Tongue Cancer*”.

Comment #2: Abstract says: ‘Next, using an orthotopic tongue cancer mouse model, we show that loss of MMP10 or miR-944 overexpression suppresses proliferation, migration, and invasion of tongue cancer cells impeding nodal metastasis.’ This sentence should be changed as it gives an impression that all this was shown by mouse model. Instead, mouse model only provided data for the role of MMP10 and *in vitro* evidence (or partly human samples) supported the rest.

Response: The abstract is revised and written as “...*We demonstrate that proliferation, migration, and invasion of tongue cancer cells are suppressed by MMP10 knockdown or miR-944 overexpression. Next, we show that depletion of MMP10 prevents nodal metastases using an orthotopic tongue cancer mice model...*”.

As an additional set of evidence to suggest sufficiency of *MMP10* overexpression to promote metastases, we performed additional *in vivo* experiment of *MMP10* overexpressing AW13516 cells injected orthotopically in the tongue of NOD-SCID mice and screened for metastasis. Interestingly, *MMP10* overexpression is sufficient to promote nodal and distant metastasis *in vivo*. We have included the data in **Figure 5, A-B** and **Supplementary Figure 13A** in the revised manuscript (as shown below).

Figure 5 (A-B): *MMP10* promotes metastases of tongue cancer *in vivo*. (A) IVIS imaging for detection of metastasis in mice injected ($n=6$ /group) with AW13516 cells stably overexpressing empty vector or *MMP10*. Bioluminescence imaging was performed after the resection of primary tongue tumors from the mice. The black arrowhead indicates the metastasis in regional lymph nodes and distant organs. (B) Tabular representation of the number of cells injected orthotopically into the tongue of mice and number of mice with metastasis in regional lymph nodes or distant organs.

Supplementary Figure S13 (A): IVIS imaging of mice organs showing metastasis *in vivo*. IVIS imaging of mice internal organs (sub-maxillary glands and cervical lymph nodes (M), lungs (L), liver (Li), kidney (K), spleen (S), uterus (U) and ovaries (O)) after necropsy to detect metastasis in mice injected with AW13516-vector control and MMP10-overexpression clones. Bioluminescence signal from the organs indicate metastasis.

The results are mentioned in the abstract as, “...overexpression of *MMP10* leads to opposite effects upregulating epithelial-mesenchymal-transition, mediated by a tyrosine kinase gene, *AXL*, to promote nodal and distant metastasis *in vivo*...”

Also, the experiment is detailed and merged with the earlier results of orthotopic tongue tumor study with *MMP10* knockdown clones of AW8507 and CAL27 in the results section of the revised manuscript as, “...we established orthotopic tongue tumor mouse model using the *MMP10* overexpression and knockdown clones. We injected luciferase-tagged AW13516 clones of *MMP10* overexpression, and CAL27 and AW8507 clones of *MMP10* knockdown (sh-2) along with vector control cells orthotopically into the tongue of 6 mice/group using needle gauge (30 G) after anesthetizing the mice with isoflurane. Caliper measurements at regular intervals showed that AW13516 cells form tumors after 25 days, CAL27 cells form tumors after 10 days, whereas, AW8507 cells formed tumors after 35 days of injection. However, the volume of AW8507-induced tumors was higher than CAL27-induced tumors, followed by AW13516-induced tumors. Interestingly, knockdown of *MMP10* significantly reduced the tumor volume in both AW8507-induced and CAL27-induced compared to the control group (Supplementary Figure S12, A-B), suggesting the role of *MMP10* in tumorigenesis of tongue cancer. Also, AW13516- and AW8507-induced tumors metastasized to cervical lymph nodes, lung, liver and kidney whereas, CAL27-induced tumors did not metastasize to any organ or lymph nodes (Supplementary Figure S12C, S13, A-C). Of note, the CAL27 cell line is reported to be a non-metastatic cell line in literature as well (Ji, J. *et al. International journal of clinical and experimental pathology*, 2018). In AW13516-induced tumors, metastasis was observed in 6/6 (100%) mice in the *MMP10* overexpression group and 1/6 (16.7%) in vector control group. In AW8507-induced tumors, metastasis was observed in 6/6 (100%) mice in the control group and none (0/6) in the *MMP10* knockdown group (Figure 5, A-D), suggesting a significant increase in the metastasis of tongue cancer upon overexpression of *MMP10* and decrease in metastasis upon depletion of *MMP10*...”

Comment #3: Page 15. row 346: ...'regulates lymph node metastasis.' please tone down as the conclusion is too strong based on the data.

Response: Following the reviewer's suggestion, the sentence in the results section is now revised as “...suggesting that *miR-944/MMP10* is potentially involved in regulating lymph node metastasis...”.

Comment #4: Conclusion that AXL is connected to miR-944 cannot be made solely based on the connection of MMP10 with miR-944 and MMP10 with AXL separately (abstract, results p15. row. 348 title)

Response: We have modified the text in the abstract section as “...*AXL* expression is essential and sufficient to mediate the functional consequence of *MMP10* overexpression...”

Also, the title of the result section is revised to “...*MMP10* driven invasion and migration of tongue cancer cells is mediated by the *AXL* signaling pathway...”.

Comment #5: Page 3. row 70: lymphogenesis to lymphangiogenesis. Please correct.

Response: We apologise for the typo. We have now corrected the typographical error from “lymphogenesis” to “lymphangiogenesis”.

Comment #6: ‘N-zero clinical trial’, please provide short explanation for this.

Response: An important prognostic factor for oral cancer is lymph node metastases. Due to occult lymph node metastases, which affects about 30% of patients with early-stage (T1 and T2) oral cancer, there are conflicting treatment recommendations. One method is elective neck dissection (END) which involve surgical excision of lymph nodes together with the original tumor, and the other is watchful waiting for patients who develop nodal metastases during follow-up therapeutic neck dissection (TND). On the basis of 500 patients with early-stage (T1 and T2) oral cancer, the N-zero clinical trial compared the survival advantages of END and TND. The study established, despite higher cost and higher morbidity associated, END as a standard of care for the early-stage clinically node negative (T1, T2 and N0) oral cancer patients.

Based on reviewer’s suggestion, we have included a short explanation on N-zero clinical trial in the results section of revised manuscript as “...*On the basis of 500 patients with early-stage (T1 and T2) oral cancer, the N-zero clinical trial compared the survival advantages of END and TND. The study established END as a standard of care for patients with early-stage clinically nodal negative (T1, T2 and N0) oral cancer and showed its superiority in terms of overall and disease-free survival rates....*”

Comment #7: Page 20, row 453. ‘Anti-correlate’ maybe other expression should be used?

Response: The sentence is changed in revised manuscript as “...*miR-944* expression is significantly negatively correlated with *MMP10* expression...”.

Comment #8: Supplement SFigure3 is mentioned before SFigure2. Please check for the correct order.

Response: We apologize for the error. The order of all the figures is now updated as per citation in the results section of the revised manuscript.

Comment #9: Did the authors consider other methods to study proliferation as passaging and counting by hemacytometer does not accurately measure proliferation (could be mix of differences in viability, proliferation, apoptosis) and is prone to errors. Did the authors test differences in apoptosis separately?

Response: We thank the reviewer for this query. We have now performed additional experiments to address the reviewer's query. In brief, we performed cell viability and apoptosis experiments with MMP10 knockdown clones of AW8507 and CAL27 cell lines, along with a western analysis to check the levels of Caspase-3 and PARP cleavage. We were unable to conduct an apoptotic experiment using Annexin V-FITC labelling because the shRNAs of the *MMP10* knockdown clones of AW8507 and CAL27 were GFP-tagged. As a result, we used AW8507 and CAL27 cells that had MMP10 knocked down using siRNA for the experiment. The data is now included in the manuscript as Supplementary Figure S3 (as shown below with the legend):

Supplementary Figure S3: Knockdown of MMP10 does not affect apoptosis of tongue cancer cells. (A and E) Bar plot of cell viability assay performed using Propidium Iodide (PI) staining indicates the percentage of late apoptotic and necrotic cells in MMP10 knockdown clones (sh-1, sh-2 and sh-3) and non-targeting shRNA control clones (sh-NT) of AW8507 (A) and CAL27 (E). (Bottom Western blots) Western blots indicate the expression of Caspase 3 and PARP cleavage in the MMP10 knockdown and vector control cells. β -actin or Tubulin was used as loading control. (B and F) qRT-PCR analysis indicating siRNA mediated knockdown of MMP10 in AW8507 (B) and CAL27 (F) cells. GAPDH was used for normalization. (C and G) Bar plot of Annexin V-FITC/PI staining indicating percentage of early- and late apoptotic cells in MMP10-knockdown and scrambled control clones of AW8507 (C) and CAL27 (G). (D and H) Representative scatterplots depicting PI (y-axis) vs. annexin V-FITC (x-axis) staining in MMP10 knockdown and scrambled control. In each scatterplot, top-left quadrant indicates necrotic cells, top-right quadrant indicates late apoptotic cells, bottom-left quadrant indicates live cells and bottom-right quadrant indicates early apoptotic cells. Data are shown as means \pm SD. *p*-values are from Student's unpaired *t*-test and denoted as ns (not significant); ****, $p < 0.0001$. Data shown are representative of three independent experiments.

The protocol followed is mentioned in the materials and methods section as:

“...Apoptosis assay

Apoptosis was measured by double-labeling the cells with Annexin V-fluorescein isothiocyanate (FITC) and propidium iodide using FITC Annexin V Apoptosis Detection Kit, as per the manufacturer's instructions (Cat No. 556570, BD Pharmingen). Briefly, 0.3 million cells of AW8507- and CAL27-MMP10 knockdown along with scrambled control cells were re-suspended in 100 μ l 1X Annexin V binding buffer and incubated on ice with 2 μ l Annexin V-FITC conjugate for 15 min. This suspension was diluted to a final volume of 300 μ l using ice-cold 1X Annexin V binding buffer and acquired immediately on AttuneNxt flow cytometer after addition of 1 μ l Propidium iodide (PI). Data analysis was performed using FlowJo software. All the experiments including siRNA-mediated knockdown of MMP10 in both cell lines were performed three times.

Cell viability assay

Cell viability was assessed by resuspension of 0.3 million cells in 300 μ l 1X Annexin V binding buffer and addition of 1 μ l of 100 μ g/ml Propidium iodide (PI). Percent viable and late apoptotic/necrotic cells were acquired on AttuneNXT flow cytometer. Data analysis was performed using FlowJo software. All the experiments were performed in three replicates..."

Also, the data is explained in the results section as:

"...To check if difference in proliferation rate is due to the difference in viability or proliferation or apoptosis. Using propidium iodide (PI) staining and flow cytometer-based sorting of live and dead cells, we conducted a cell viability study with shRNA-mediated *MMP10* knockdown clones of AW8507 and CAL27. Results did not show significant difference in the viability of cells upon *MMP10* knockdown (Supplementary Figure S3, A, E). The results were further supported by immunoblotting, which revealed comparable amounts of caspase-3 and PARP cleavage in the *MMP10* knockdown clones of AW8507 and CAL27 compared to control cells (Supplementary Figure S3, A, E). We performed siRNA-mediated knockdown of *MMP10* in AW8507 and CAL27 cells. The knockdown of *MMP10* was confirmed using real-time PCR (Supplementary Figure S3, B, F). Similar to viability assay, Annexin V-FITC/PI staining did not show significant difference in apoptosis of cells upon knockdown of *MMP10* (Supplementary Figure S3, C-D, G-H). These findings demonstrate that the difference in cell proliferation rates following *MMP10* knockdown is what causes the reduction in cell proliferation rates..."

Comment #10: Some methods properly describe 1) the number of individual experiments performed and 2) number of technical repeats in each experiment. However, some lack this information. Please add this information for all experiments. Without this information, it is impossible to evaluate the results.

Response: We agree with the reviewer. We have now mentioned the number of technical repeats performed for each experiment and number of times each experiment was performed in the materials and methods section and figure legends (Figure 1, 2, 3, 4; Supplementary Figures S2, S3, S4, S5, S6, S7, S8, S10, S11) of the revised manuscript.

Comment #11: There is some uncertainty (partly because of the above-mentioned shortages on repetitions of experiments) on how statistical analyses were performed in each experiment. For example, in Figure 2, many of the Figures show relative values to control (=1 or 100). From which values were the P values calculated? It would be more informative to illustrate the values used for statistical analyses in Figures and not the ratios to control. Also, for 2E, 2I, the statistics are confusing. Are the controls (values from technical repeats, individual experiments?) compared to all sh cells? The differences illustrated by these experiments seems clear, but statistics should be performed properly and described thoroughly. Without this information, it is impossible to evaluate the results.

Response: We concur with the reviewer's comment. We have now clarified in the methods section and figure legends of the text how many times each experiment was conducted in order to allay the issue. All these experiments were performed in three replicates. As suggested by the reviewer, we have re-plotted all the invasion and wound healing experiment graphs and determined the *p*-values with the actual experimental values in Figure 2 (Figure 2, B–C, F–G, J–K) using unpaired student's *t*-test. Also, along with Figure 2, we have now re-plotted all the migration and invasion assay graphs in Figure 3, E–F, Figure 4, F–K, of the revised manuscript using actual experimental values. Earlier, we used relative values to control (=1 or 100) experimental data for plotting the graphs and deriving the *p*-values. For the cell proliferation assay (Figure 2E, 2I), the *p*-values were determined by contrasting

the control (sh-NT) with each of the three sh-MMP10 clones (sh-1, sh-2, and sh-3), while keeping the p -values determined using the actual experimental values or relative values to the control constant. We have now plotted the relative values to control after taking into account the reviewer's comment. We calculated and noted the p -values for each sh-MMP10 clone (sh-1, sh-2, and sh-3) compared to sh-NT control in the revised manuscript using Day 6/Day 1 ratio values.

Comment #12: Do the authors have reason to use P values <0.005 for ** and <0.0005 for *** as more commonly used values would be <0.01 and <0.001.

Response: We agree with the reviewer's comments. The p -value range for *, ** and *** has been corrected in the manuscript figures and confirmed the same in the statistical analyses using GraphPad prism 8. The corrected range for p -values mentioned in the manuscript are as follows: *, $p < 0.05$; **, $p < 0.01$; ***, $p < 0.001$; ****, $p < 0.0001$.

Comment #13: For Western blots, molecular weight standards should be marked (for all) figures and original, whole blot images should be provided as Supplementary data. Without this information, it is impossible to evaluate the results.

Response: We fully concur. We have marked the molecular weight standards for all Western blots in the revised manuscript figures (Figure 2 (A, D, H), Figure 4 (A-B), Supplementary Figure S2C, S3A, S3E, S7D, S10B) and also provided the whole blot images as Supplementary Figures S14-S25.

Comment #14: How many times the blots were performed? For at least most important differences claimed based on the results, three blots should be performed and the differences in band intensities measured and related to housekeeping protein.

Response: We agree with the reviewer. Three independent replicates of each Western blot were performed, in all our experiments. Using ImageJ software, we have now quantified the band intensities and compared them to the reference control. In the Supplementary Figures S14-S25, the quantified band intensities are shown as values underneath the original whole blot photographs.

Comment #15: In Figure 2, two separate blots for vinculin as housekeeping are shown. However, twelve proteins were detected and presumably, they are not all from two different blots with vinculin in addition. If this is the case, housekeeping should be provided for all blots separately. Without this information, it is impossible to evaluate the results.

Response: In response to a reviewer's recommendation, we have individually added housekeeping genes for the blots in the updated Figure 4, A-B of the revised manuscript.

Comment #16: IHC: the authors do not provide information for the antibody used? Should there be also isotype control to show the antibody specificity – that depends on the antibody used. Without this information, it is impossible to evaluate the results.

Response: We agree with the reviewer. The information on the MMP10 antibody (Human MMP10 Monoclonal Antibody (Cat No. MAB910, Biotech)) utilised for the IHC experiment has been updated in the materials and methods section of the revised manuscript as "...The slides were blocked by horse serum for 1 h before incubating with the MMP10 primary antibody (Human MMP10 monoclonal antibody (Cat No. MAB910, Biotech)) overnight at 4°C in a moist chamber...". To demonstrate the antibody specificity, we included primary antibody-only and secondary antibody-only controls in the experiment (Supplementary Figure S1, A-B), as shown below in the figure in Comment #17 of Reviewer #1.

Comment #17: The authors should describe in the text (and for this, please provide higher magnification images along with the current ones) where is the MMP10 IHC staining localized – is it inside the cells, on membrane, extracellular? Also, the distribution of MMP10 in tumors would be highly informative and interesting – e.g. is it expressed in the invasive front, close to certain tissue structures etc.?

Response: Using IHC, we detected MMP10 staining in the cytoplasm of tumour cells. Compared to the centre of the tumour, the invasive front of tumours had higher expression levels of MMP10. As suggested by the reviewer, we have added the IHC localization data as Figure 1D, in the revised manuscript (as shown in the figure below).

Accordingly, we have updated the results section of the manuscript as “...MMP10 protein was mainly detected in the cytoplasm of tumor cells (Figure 1D). IHC results suggest significant overexpression of MMP10 protein in tongue cancer patients with lymph node metastasis ($p=0.0075$) (Figure 1A, 1C, and Supplementary Figure S1C). Additionally, we find increased expression of MMP10 at the invasive fronts of the tumor compared to the centre of the tumor (Figure 1D)...”

Figure 1D: Immunohistochemistry of MMP10 in N-zero clinical trial samples. (D) Representative IHC stained images of tongue tumors are shown with a scale bar. The brown color indicates positive staining for MMP10 protein. (Upper panel) IHC stained images showing expression of MMP10 in cytoplasm of tumor cells. (Lower panel) IHC stained images showing the expression of MMP10 in the middle region of tumor and at the invasive ends of tumor.

Comment #18: There are some errors in reference list, e.g. Ramqvist, T. et al. MMP-10/Stromelysin-2 Promotes Invasion of Head and Neck Cancer. PLoS ONE 539 6, e25438, doi:10.1371/journal.pone.0025438 (2011). The first author is incorrect.

Response: We apologize for the error. The reference is corrected and updated in the revised manuscript as: “...13. Deraz, E. M. et al. MMP-10/Stromelysin-2 Promotes Invasion of Head and Neck Cancer. PLoS ONE 6, e25438, doi:10.1371/journal.pone.0025438 (2011)...”

Reviewer #2

Comment #1: In Figure 1, authors showed that MMP10 overexpression was involved in tongue cancer patients with lymph node metastasis. However, authors previously reported that MMP10 overexpression was associated with lymph node metastasis in early tongue cancer patients. Even if

the tumor size is small (T1 and T2), if patients have metastasis, authors cannot say early stage. Authors should not use the term “early stage”. Moreover, authors should explain why you focused on T1 and T2 cancer. Moreover, in your previous study, authors used HPV negative early-stage tongue cancer patients habitual of chewing betel nuts, areca nuts, lime or tobacco. In general, cancer patients habitual of chewing betel nuts, areca nuts, lime or tobacco have buccal cancer. Why authors focused on tongue cancer in previous and current studies?

Response: Based on the AJCC (American Joint Committee on Cancer)/ UICC (Union for International Cancer Control) TNM classification system (8th edition) guidelines, primary oral cavity tumors are clinically staged as T1 (measuring ≤ 2 cm) and T2 (measuring >2 cm but <4 cm), and are classified as early-stage. The occult lymph node metastasis present in these patients is not detected by physical examination or imaging techniques like PET/CT or ultrasonography, hence, these patients are termed as clinically early-stage oral cancer patients. As suggested by the reviewer, we have now included AJCC/UICC guidelines in the materials and methods section of the revised manuscript as “...Primary tongue tumors were staged as T1 (measuring ≤ 2 cm) or T2 (measuring >2 cm but <4 cm) as per AJCC (American Joint Committee on Cancer)/ UICC (Union for International Cancer Control) TNM classification system (8th edition) and referred as clinically early-stage tongue tumors...”

We focused on T1 and T2 stage of tongue cancer since these individuals are known to have occult lymph node metastasis, which reduces their chance of survival by 50% when compared to those without lymph node metastases.

We concur with the reviewer that individuals who regularly chew betel, areca, lime, or tobacco have an increased risk of developing tongue, buccal, and other oral cavity subsites cancer. It's interesting to note that tongue cancer is the most prevalent subtype in industrialised nations and is known to affect younger people more frequently in India (Sherin N. *et al.*, Indian J. Cancer, 2008). Thus, the subject of our research was tongue cancer. 60% of patients with tongue cancer in our study chew tobacco, which is consistent with the literature and the reviewer's comments, as mentioned in the introduction.

Comment #2: Authors should demonstrate that the involvement of miR-944/MMP10/AXL-axis is specific in T1 and T2 tongue cancer. To prove your hypothesis, authors should explain the following question.

- 1) How about the role of miR-944/MMP10/AXL-axis in T3 and T4 tumors?
- 2) How about the involvement of miR-944/MMP10/AXL-axis in early-stage cancer (T1-T3 without metastasis)?
- 3) How about the role of miR-944/MMP10/AXL-axis in the precancerous lesions (ex. epithelial dysplasia)?
- 4) How about the role of miR-944/MMP10/AXL-axis in HPV-positive cancers?

Response: The reviewer's suggestion to look into additional cancer stages are outstanding and an ongoing query in our laboratory. These elements, however, are outside the purview of the current manuscript and being investigated independently.

Comment #3: Authors should examine the MMP10 overexpression and check the upregulation of AXL. In addition, authors should check the phenotype of AXL overexpression.

Response: In response to the reviewer's suggestions, we performed additional *in vitro* cell-based experiments and overexpressed AXL in AW1516 and the MMP10 knockdown clone (sh-2) of AW8507. The revised manuscript includes the assay results (Figure 4, A-K), which are described in the results section as: “...To study if AXL regulates MMP10-mediated proliferation, migration, and invasion of cells in the downstream, AXL overexpression and knockdown clones were used for *in vitro* metastasis-related cell-based assays. AXL knockdown significantly reduced the biochemical levels of p-AKT, p-

MTOR and p-NFKB levels and impeded the *MMP10*-driven proliferation, migration, and invasion phenotypes, whereas *AXL* overexpression clones restored the p-AKT and p-NFKB levels along with proliferation, migration, and invasion phenotypes of *MMP10*-knocked down AW8507 cells (Figure 4, A-K). These findings suggest that *AXL* is downstream of *MMP10* and mediates the metastatic properties of tongue cancer cell lines upon upregulation by *MMP10*...”

Figure 4: miRNA-944/MMP10 induced EMT is mediated by the AXL signalling pathway. (A-B) Immunoblot of *MMP10*, p-*AXL*, *AXL*, p-*AKT*, *AKT*, p-*mTOR*, *mTOR*, p-*NF-κB*, *NF-κB*, Vinculin and β-actin upon overexpression/knockdown of *MMP10* or overexpression/knockdown of *AXL* in tongue cancer cell lines. Vinculin or β-actin was used as loading control. Four Western blots probed for Vinculin are labelled with “v” symbol on the blot in the panels. For all other immunoblots in the panel, β-actin was used as loading control. (C-E) Cell proliferation assay of AW13516 cells stably overexpressing empty vector or *AXL* (C), AW8507-MMP10 KD cells stably overexpressing empty vector or *AXL* (D) and AW13516-MMP10 OE cells with scrambled siRNA (si-scr) or siRNA-mediated knockdown of *AXL* (E). Scatter plots indicate the total number of live cells on the mentioned day. (F, H and J) Boyden chamber migration and (G, I and K) invasion assays of AW13516 cells stably overexpressing empty vector or *AXL* (F and G), AW8507-MMP10 KD cells stably overexpressing empty vector or *AXL* (H and I) and AW13516-MMP10 OE cells with scrambled siRNA (si-scr) or siRNA-mediated knockdown of *AXL* (J and K) plotted as bar plots indicating the percentage of cells migrated or invaded, respectively. Representative images of crystal violet stained Boyden chambers. Data are shown as means ± SD. *p*-values are from Student's unpaired *t*-test and denoted as ns (not significant); *, *p* < 0.05; **, *p* < 0.01; ***, *p* < 0.001. Data shown are representative of three independent experiments.

Comment #4: Authors showed that *MMP10* overexpression promotes proliferation, migration, and invasion via *AXL* overexpression. How about the mechanism (how *AXL* regulates proliferation, migration, and invasion)?

Response: We thank the reviewer for this pertinent query. To examine the activation of *AXL* pathway genes (p-*AKT*, p-*mTOR*, and p-*NF-κB*) affecting proliferation, migration, and invasion in tongue cancer cell lines, we generated additional reagents and performed experiments using these *AXL* overexpressing and knockdown clones. The results of the western blotting used to check for the

expression of these AXL downstream genes are described in the results section and are included in the figures (Figure 4B, as shown above in response to Comment #3 of the Reviewer #2) as: “...we investigated whether AXL overexpression in MMP10 knockdown clones or AXL knockdown in MMP10 overexpression clones would reverse the activation or inactivation of AXL downstream signaling, respectively. We performed stable overexpression of AXL in AW13516 (cell line with endogenously low levels of MMP10) and AW8507-MMP10 knockdown cells, and siRNA-mediated knockdown of AXL in AW13516-MMP10 overexpression clones. We found that overexpressing AXL activates AXL signalling in MMP10 knockdown cells whereas knocking down AXL inactivates AXL signalling in MMP10 overexpressing cells (Figure 4B), indicating that AXL signalling is downstream of MMP10...”

Reviewer #3

Comment #1: The main finding in the paper relies on the proposed regulatory mechanism of miR-944/MMP10/Axl axis. In this regard, it is not clear why the authors did not evaluate the basal expression level of Axl in all cell lines employed in this study. For instance, only the levels of MMP10 were demonstrated (Figure S2). However, the authors did not show the levels of Axl in the cell. Does Axl levels correlate with MMP10 levels in all cell lines? And in tissue samples?

Response: Following the reviewer's comments, we performed real-time PCR and Western analysis to screen for the expression of AXL transcript and protein in the tongue cancer cell lines (AW13516, AW8507 and CAL27). The data has been included in the Supplementary Figure S10, A-B, of the revised manuscript (as shown below). We had also screened for the expression of AXL transcript in tongue tumor samples (N=52) (Supplementary Figure S8 in the revised manuscript). Although the expression of AXL does not correlate with MMP10 expression in the cell lines, possibly due to limited number of cell lines, it shows significant correlation ($p < 0.01$) in tissue samples.

Supplementary Figure S10: Expression of AXL transcript and protein in tongue cancer cell lines. (A) qRT-PCR analysis for estimation of AXL in tongue cancer cell lines. Expression of AXL was normalized with GAPDH. Data are shown as means \pm SD. (B) Immunoblot showing expression of p-AXL, AXL and β -actin protein in tongue cancer cell lines. Data shown are representative of three independent experiments.

Comment #2: What are the activation status (phosphorylated form) of Axl in the cell lines? And in the tissue samples?

Response: Following the reviewer's query, we have performed Western blotting to screen for the expression of phosphorylated AXL in the TSCC cell lines (AW13516, AW8507 and CAL27). The data is included in the Supplementary Figure S10 of revised manuscript (as shown above in response to Comment #1 of the Reviewer #3). Screening for phosphorylated AXL in tissue samples would have been highly informative, however, we could not assess the expression due to limiting tissue samples.

Comment #3: The mechanistic analysis was performed in MMP10 overexpressed cells. Considering that AW8507 and CAL27 express high levels of MMP10, why the authors overexpress this protein in a cell line and did not evaluate the mechanism in AW8507 and/or CAL27?

Response: By knocking down *MMP10*, we have assessed the mechanistic analysis in the AW8507 and CAL27 cell lines. When *MMP10* is silenced, we see the deactivation of the AXL signalling pathway. It's interesting to note that the AXL signalling pathway was reactivated when we overexpressed *AXL* in the *MMP10* knockdown clones of AW8507 (sh-2). The results are further explained and covered in response to the comment #4 of the Reviewer #3 (below).

Comment #4: Since the authors are claiming that MMP10 induces Axl expression, it would be important to demonstrate whether MMP10 indeed modulates Axl and proteins in the Axl pathway expression. In this regard, would be interesting to use MMP10 knocked down and evaluate Axl and Axl pathway protein levels

Response: We agree with the reviewer. In response to the query, we have evaluated AXL and AXL pathway proteins in the *MMP10* knockdown clones of AW8507 and CAL27 clones. The western blotting results are included in Figure 4A (as shown above in response to Comment #3 of the Reviewer #2). We have detailed the experiment in the results section as “...we screened for the expression of AXL downstream signaling molecules (AKT, mTOR and NF- κ B) upon overexpression and knockdown of *MMP10*. Results suggest that overexpression of *MMP10* upregulates AXL and activates downstream signaling pathway whereas stable knockdown of *MMP10* suppresses these effects (Figure 4A)...”

Comment #5: There is lack of consistency in the western blots. MMP10 expression levels are completely different for cell line AW13516 in figure 3I and figure S2C (Levels of MMP10 in figure S2 is different when compared to Figure 3I). Why did the authors used both GAPDH (figure S2C) and vinculin (figure 3I) as loading control?

Response: We thank the reviewer for giving us an opportunity to clarify. Due to a discrepancy in exposure duration during blot creation in ChemiDoc, the MMP10 protein levels for the AW13516 cell line as shown in the western blotting data of Figure 3I (updated as Figure 4A in the revised manuscript) and Figure S2C were different. Additionally, there were differences in the amounts of protein loaded in the two gels (40 μ g was loaded in the Western blot of Figure 3I versus 50 μ g in Figure S2C). For better representation, we've replaced the western blot in Figure 3I (updated Figure 4A) with a comparable exposure blot loaded with the same quantity (50 μ g) of protein.

To eliminate molecular weight overlap with the loading control protein on the same blot, we employ GAPDH (35 kDa), Vinculin (124 kDa), α -tubulin (55 kDa) or β -actin (42 kDa) as loading controls.

Comment #6: Quantification and normalization of proteins in the Western blots protein should be performed.

Response: We have quantified the proteins in all the Western blots (Figure 2 (A, D, H), Figure 4 (A-B), Supplementary Figure S2C, S3A, S3E, S7D, S10B) and the values are presented along with raw western blot images in Supplementary Figures S14-S25.

Comment #7: A figure summarizing the regulation of miR-944/MMP10/Axl axis in this model, should be included in the manuscript.

Response: We are highly grateful to the reviewer for the query. We have now included a figure summarizing the regulation of *miR-944/MMP10/AXL*- axis in tongue cancer. The model is included as Figure 6 in the revised manuscript (as shown below):

Figure 6: Model depicting miR-944/MMP10/AXL- axis to regulate metastasis in early-stage tongue cancer. The working model demonstrating the role of MMP10 in promoting metastasis via activation of AXL signaling pathway and is negatively regulated by miR-944 in early-stage tongue cancer.

Reviewer #4

Comment #1: Please consider revising the statement on the second sentence (rows 37-38). This result is not exactly clear in results section (see comment x) nor does the given n correspond to the number of node-positive tumors.

Response: Our study includes a total of 219 early-stage tongue tumor samples. 81 samples were taken from patients positive for lymph node metastases of 219 samples. MMP10 is overexpressed in 70 of these 81 samples, or 86% of them. In response of the reviewer's comment, for clarity, the statement in abstract has been changed to read as “...Our immunohistochemical and real-time PCR analysis of 219 early-stage tongue tumors show MMP10 overexpression in 86% of node-positive tongue tumors (n=81; p<0.0001)...”.

Comment #2: The sentence starting from row 41 “Next, using orthotopic tongue cancer model...” gives the impression that all functional aspects (proliferation, migration and invasion) were studied in the mouse model, which based on the results, is not the case (they were studied in vitro). Please consider revising this sentence.

Response: We thank the reviewers for the comment. The text in the abstract is re-written as “...We demonstrate that tongue cancer cell proliferation, migration, and invasion are suppressed by MMP10 knockdown or miR-944 overexpression. Next, we show that depletion of MMP10 prevents nodal metastases using an orthotopic tongue cancer mice model...”

Comment #3: For Keywords, please consider including AXL or miRNAs instead of “Epithelial mesenchymal transition” as they could better reflect the results and conclusions of this manuscript.

Response: We have updated the list of keywords with “AXL” and “miR-944” instead of “Epithelial mesenchymal transition”.

Comment #4: LNM is indeed found to result in worse prognosis for OTSCC patients, as suggested by authors in first part of the introduction (rows 60-62). Yet the references used do not discuss OTSCC but other cancers, although such research and associated literature exists. Please consider referring to OTSCC-related publications or modifying the sentence accordingly.

Response: We apologize for the overlook on our part. We have updated following references indicating poor prognosis of OTSCC patients with lymph node metastasis in the introduction section of the revised manuscript.

“...4. Yang, W. *et al.* Lingual Lymph Node Metastasis in cT1-2N0 Tongue Squamous Cell Carcinoma: Is It an Indicator for Elective Neck Dissection. *Frontiers in Oncology* **10**, doi:10.3389/fonc.2020.00471 (2020).

5. Kuroshima, T. *et al.* Prognostic impact of lingual lymph node metastasis in patients with squamous cell carcinoma of the tongue: a retrospective study. *Scientific Reports* **11**, doi:10.1038/s41598-021-99925-2 (2021).

6. Larsen, S. R., Johansen, J., Sørensen, J. A. & Krogdahl, A. The prognostic significance of histological features in oral squamous cell carcinoma. *Journal of Oral Pathology & Medicine* **38**, 657-662, doi:10.1111/j.1600-0714.2009.00797.x (2009)...”

Comment #5: Row 70: Please consider the term “lymphogenesis”, should it read “lymphangiogenesis” instead?

Response: We thank the reviewer for the correction. We have now corrected the typographical error from “lymphogenesis” to “lymphangiogenesis”.

Comment #6: The sentence on rows 74-76 only refers to ovarian cancer, although he referenced studies also include HNSCC and cervical cancer. Please consider revising this sentence.

Response: We have revised the sentence in the introduction section as “...**Overexpression of MMP10 has been reported to promote invasion, metastasis and regulate stemness of cancer cells through activation of Wnt signaling pathway in head and neck, and ovarian cancer, and promote tumor progression by regulating angiogenic and apoptotic pathways in cervical cancer...**”

Comment #7: Please consider toning down the conclusion in the sentence on rows 79-81. Even if MMP10 have been upregulated in another head and neck cancer and relates to pre-metastatic niche formation in cancers overall, the variation between cancers overall and within HNSCCs is great. Thus, it is hard to convincingly suggest that MMP10 would play a LNM-related role in OTSCC based on these factors, although it warrants the research on this topic.

Response: We agree and thank the reviewer for the suggestion. We have revised the text in the introduction as “...**Additionally, MMP10 has been reported to promote pre-metastatic niche formation and found to be upregulated in early-stage esophageal cancer patients. However, the role of MMP10 remains unexplored in lymph node metastasis in early-stage tongue cancer...**”

Comment #8.1. This study focuses on tongue cancer, yet the patient cohorts also include tumors from the anatomical site of buccal mucosa. Could the authors explain why and describe, whether these tumors were excluded from the analyses or not?

Response: Based on reviewer’s suggestion, we performed additional analyses by excluding buccal mucosa samples-- as presented in the graphs below (for reviewer’s information only) — (A) graph showing the results of MMP10 immunohistochemistry, and (B) graph showing the delta CT values corresponding to the *MMP10* transcript expression compared between node positive and node negative tongue tumor samples. These graphs are consistent with the graphs in Figures 1A and 1B (analysed by inclusion of buccal mucosa samples)-- with no significant difference when buccal mucosa

samples were included or excluded from the analyses. Moreover, the buccal mucosa patient samples were collected retrospectively from the N-zero clinical trial samples, we thus retained the buccal mucosa samples in the study.

Figure (A-B): MMP10 overexpression is associated with nodal metastasis in early-stage tongue cancer patients. (A) IHC of MMP10 in early-stage primary tongue tumors (N=98). One dot represents IHC score of MMP10 in one sample. (B) qRT-PCR of MMP10 transcript expression in early-stage primary tongue tumor samples (N=110). GAPDH was used as reference control. Data is plotted as box plot representation of delta Ct (dCT) values. p-values are from Mann-Whitney tests and denoted as ***, $p < 0.001$. Data shown are representative of three independent replicates.

Comment #8.2. Could the authors describe what is the difference between smoking and tobacco status?

Response: The smoking status in our study refers to patients with habits of smoking tobacco as cigarettes, cigars or pipes whereas tobacco status represent the patients with the habit of smoke-less tobacco chewing. Based on reviewer’s suggestion, we have included following text in the materials and methods section of the revised manuscript as: “...The patient details include age, gender, anatomic site, TNM tumor stage, nodal status, smoking (represents the patients with habits of smoking tobacco as cigarettes, cigars or pipes), chewing, alcohol, tobacco (represents the patients with the habit of smoke-less tobacco chewing), recurrence, metastasis and status at last follow-up...”

Comment #8.3. What was the last follow-up date of these patients? Did it vary or were all followed to the suggested 5 years mark?

Response: The last follow-up date of these patients ranges from 1 month to 11.7 years with a median follow-up of 3.9 years. As the patient samples were collected retrospectively, the follow-up period was not reached to 5-year mark in all patients. We have included this information in the materials and methods section of the revised manuscript as: “...status at last follow-up (the last follow-up date of patients ranges from 1 month to 11.7 years with a median follow-up of 3.9 years. As the patient samples were collected retrospectively, the follow-up period was not reached to 5-year mark in all patients)...”

Comment #9: For western blotting (page 6 in M&M, page1-2 in supplementary M&M): How was the unspecific antibody binding blocked?

Response: We have included the information about blocking step in materials and methods section as “...PVDF membrane blocking was performed using 5% BSA to avoid nonspecific binding of antibody...”

Comment #10: In general, the methods describe both technical replicates in assays as well as number of repeated experiments in a varying manner resulting this information to be missing in some assays

(e.g. boyden chamber assays). Could the authors please provide this information for all the assays when necessary (this suggestion includes supplementary M&M).

Response: We have included the details regarding the number of times each experiment was repeated in the materials and methods section and figure legends (Figure 1, 2, 3, 4; Supplementary Figures S2, S3, S4, S5, S6, S7, S9).

Comment #11: For figure 1B: Could the authors explain why they have presented the gene expression values as negative delta Ct values? This is quite an uncommon presentation.

Response: The gene expression values were presented as negative delta Ct values for better visual comparison of gene expression with the variable in comparison. Following the reviewers suggestion, we have re-plotted the gene expression values as delta Ct values instead of negative delta Ct values in the figures (Figure 1B, Figure 3, G-H, Supplementary Figure S8) of the revised manuscript.

Figure 1B: Quantitative real-time PCR (qRT-PCR) of MMP10 transcript expression in early-stage primary tongue tumor samples (N=111). GAPDH was used as reference control. Data is plotted as box plot representation of dCT values. p-values are from Student's unpaired t-test and denoted as ***, $p < 0.001$. Data shown are representative of three independent replicates of real-time PCR data for each sample.

Figure 3, G-H: (G) qRT-PCR of miR-944 mRNA in 63 primary tongue tumor samples correlated with MMP10 mRNA expression. (H) qRT-PCR miR-944 mRNA in 93 primary tongue tumor samples correlated with nodal metastasis status of patients. Data are plotted as box plot representation of dCT values. MMP10 and miR-944 expression are normalized with GAPDH and U6, respectively. p-values are from Student's unpaired t-test and denoted as *, $p < 0.05$; **, $p < 0.01$. Data shown are representative of three independent replicates of real-time PCR data for each sample.

Supplementary Figure S8: Validation of metastasis pathway-related genes and correlation with MMP10 expression in early-stage tongue tumor samples. qRT-PCR analysis of AXL, CDH11, COL1A1, FGFBP1, IL6, IL7R, IL11, and MMP2 transcript expression in early-stage primary tongue tumor samples (N=52). Box plot representation of dCT values and their significance between tumor samples with low (MMP10 low) and high (MMP10 high) expression of MMP10 (based on median expression). GAPDH was used for normalization. Data are shown as means \pm SD. p-value was calculated using Student's unpaired t-test and denoted as ns (not significant); *, $p < 0.05$; **, $p < 0.01$; ***, $p < 0.001$. Data shown are representative of three independent replicates of real-time PCR data for each sample.

Comment #12: In the sentence on rows 251-254 authors claim OTSCC patients with LNM show higher gene and protein MMP10 expression, but seemingly the data is not provided for RT-PCR analysis. Please check.

Response: Following reviewer's suggestion, the real-time PCR and IHC data are detailed in the results section of the revised manuscript as, "...Specifically, real-time PCR and IHC data reveal MMP10 overexpression in 59/67 (88%) and 11/14 (79%), and low expression in 8/67 (12%) and 3/14 (21%) of patients with lymph node metastasis, respectively. Overall, of 219 early-stage tongue tumors we find MMP10 overexpression in 86% of node-positive tongue tumors ($n=81$; $p<0.0001$)..."

Comment #13: For Table II: Authors claim significant results in text (rows 264-265), yet the table does not provide these. Please check and if needed, modify this sentence accordingly.

Response: The sentence in the results section is re-written as "...we found a marginally significant correlation ($p<0.082$) of MMP10 transcript expression with lymph node metastasis in both analyses..."

Comment #14: For figures 2 and 3: Please provide the molecular weights (or their markers) for the western blot images (this is also a recommendation by the journal). For further validation, whole blot images could also be provided for evaluation.

Response: We have included all the Western blots with molecular weights for their markers in the figures and the whole blot images are included in the Supplementary Figures S14-S25 of the revised manuscript.

Comment #15: On page 16 authors describe the effect of high MMP10 or AXL expression on HNSC(C) patient survival. Have the authors checked if this conclusion can be made based on only OTSCC patients in this data? HNSCCs vary greatly in etiological and other factors (e.g. HPV), which might also affect survival.

Response: We have performed the survival analysis using only TCGA-OTSCC data (N=112) based on *MMP10* or *AXL* expression. Results suggest no significant difference in the overall survival (survival plots are shown below— for reviewer’s information only) based on *MMP10* (A) or *AXL* (B) expression.

Figure (A-B): Kaplan-Meier survival curves for overall survival in TCGA-OTSCC data (N=112): Kaplan-Meier survival curves for overall survival (OS) of TCGA-HNSC data based on the expression of *MMP10* (A) and *AXL* (B). The red and black lines denote the high and low expression of the genes in the KM plots. The log-rank test was used to determine the statistical differences in median survival.

Comment #16: On rows 379-381 authors claim that *AXL* could regulate metastasis, yet this conclusion is hard to justify only based on results from in vitro assays. Please consider toning down this conclusion.

Response: The text in the result sections is modified as “...indicating that *AXL* is downstream of *MMP10* and could potentially regulate the ability of tongue cancer cell lines to metastases upon *MMP10* upregulation...”

Comment #17: On row 387 authors claim to have developed the orthotopic tongue cancer mouse model when indeed this model has been introduced by Myers JN et al already in their 2002 publication in Clin Cancer Res journal. Please consider using another verb in this sentence.

Response: We have modified the sentence in the results section as “...we established orthotopic tongue tumor mouse model using the *MMP10* overexpression and knockdown clones...”.

Comment #18: Could the authors please clarify the comment on “elective neck dissection that is mandatorily performed” (juxtaposition, rows 416-419). Also please note that different countries might have different standards on elective neck dissection which should be taken into account.

Response: We have modified the sentence in the discussion section as “...While most patients with early-stage tongue cancer receive elective neck dissection as standard therapy in India and other countries, testing for *MMP10* expression may allow for the substantial majority of pathological node-negative individuals to avoid the invasive surgical operation associated with significant morbidity...”

Comment #19: Please check and possibly revise the use of term anti-correlated (unusual term, row 453) as well as consider if the sentences on rows 460-463 could be merged as they use the same references and discuss the same topic.

Response: The term anti-correlated is changed in revised manuscript and the sentence is written as “...*miR-944* expression is significantly negatively correlated with *MMP10* expression...”.

Comment #20: Reference 3 is apparently a book chapter. Please check the correct formatting for the reference type.

Response: We have updated the citation format of book chapter in the correct format in the reference section as shown below:

“...3. García-Martín, J. M. et al. Epidemiology of Oral Cancer. 81-93 (Springer Nature, 2019)...”

Comment #21: Reference 12 (Ramqvist et al. 2011) contains wrong first author.

Response: The reference is corrected and updated in the revised manuscript as shown below:

“...13. Deraz, E. M. et al. MMP-10/Stromelysin-2 Promotes Invasion of Head and Neck Cancer. PLoS ONE 6, e25438, doi:10.1371/journal.pone.0025438 (2011)...”

Comment #22: References 37 and 40 are the same reference.

Response: The duplicated reference is removed and cited once as reference 50 in the revised manuscript.

“...50. Brand, T. M. et al. AXL Is a Logical Molecular Target in Head and Neck Squamous Cell Carcinoma. Clinical Cancer Research 21, 2601-2612, doi:10.1158/1078-0432.ccr-14-2648 (2015)...”

Reviewers' comments:

Reviewer #1 (Remarks to the Author):

The authors have addressed all my concerns sufficiently and the work has improved substantially. They have performed quite a lot of new experiments to support their findings. I have minor comments that I would like to point out:

Please change the name of the protein throughout the manuscript; by NFKB you mean p65..? = NF- κ B p65

IVIS imaging method description and luciferin/substrate use in mouse and in vitro is lacking. Please add.

Please confirm that you mention in the method description that the animal experiments were performed on female mice only (if this is the case).

Reviewer #2 (Remarks to the Author):

I have read the comments from four reviewers and the response by the authors. Authors responded to the most comments. However, I think that the authors did not satisfactory responded to all comments. Therefore, I cannot recommend this paper for publication. My comments are the following:

1. The authors did not answer some critical question (Reviewer #2 comments#2). Authors should perform additional experiments to solve this question.
2. It is still unclear that authors use the term "early stage" even in the cases have lymph node metastasis. Authors described that "We focused on T1 and T2 stage of tongue cancer since these individuals are known to have occult lymph node metastasis, which reduces their chance of survival by 50% when compared to those without lymph node metastases." However, the cases with lymph node metastasis are not early stage....
3. To demonstrate the role of miR-944/MMP10/AXL axis in early stage of tongue cancer, authors used oral cancer cell lines. AW 13516 and AW 8507 cell lines derived from poorly differentiated SCC and epidermoid carcinoma of the tongue respectively were used in this study. Moreover, CAL27 cells are also used. Are these cell lines "early stage of tongue cancer"?
4. In this study, authors focus on tongue cancer and exclude buccal cancer. It is still unclear why the authors exclude buccal cancer.
5. In Figure 6, authors added the schematic model (Reviewer#3 comment#7). Downstream of AKT signaling, authors showed mTOR and NF- κ B. However, authors did not demonstrate this..

Reviewer #3 (Remarks to the Author):

The revised version of the manuscript now entitled "Role of miR-944/MMP10/AXL- axis in Lymph Node Metastasis in Early-Stage Tongue Cancer" by Dharavath et al. is considerably improved. The authors answered the questions raised in this revision step; however, two points should be considered:

1-) Although the authors provided the quantification and normalization of western blot proteins, the information is demonstrated in the supplementary material. It would be important to provide this information in the main text of the manuscript.

2-) The authors should provide the method employed for the normalization and quantification of

western blot proteins.

Reviewer #4 (Remarks to the Author):

This report concerns the revised manuscript by Dharavath et al. (COMMSBIO-22-1374A), in which the authors describe the connection between MMP10 and lymph node metastasis in OTSCC as well as its effects on AXL-signalling and as a target of miR-944. Main findings are clear and well supported by additional analyses.

The authors have sufficiently answered the concerns and questions raised by the reviewers and performed additional experiments, when needed. The quality of the manuscript has been improved in the process. The manuscript is a good fit for the scope of the Communications Biology journal and a likely interest for its readers.

Yet, there are a few issues that need to be discussed still.

1. Rev4, comment/question 8.1. (patient samples for immunohistochemistry, Fig1A): Retrospectively collected or not, buccal mucosal samples are not OTSCC and thus should A) either not be included in the analyses (especially since their exclusion does not drastically change the results of the analyses) OR B) the figure 1 legend should be modified to include information of the buccal mucosal samples included in the analyses (number of such samples in each group, title of the figure etc.).
2. Rev4, comment/question 15 (TCGA): It is evident why the authors want to emphasize the survival in HNSCC dataset instead of OTSCC dataset. Yet for transparency, it should be mentioned in either the results or discussion section that similar conclusion cannot be drawn from the TCGA-OTSCC data alone. The reasons for this discrepancy can be mentioned as well, e.g. small number of samples etc.
3. Please consider updating the information in the Reporting Summary. Information on Flow cytometry analysis is missing and potentially also of use of clinical data (unless the reporting summary specifically addresses only clinical trials).

Response to the reviewers:

We are grateful to the reviewers for their insightful comments, which we believe have allowed us to greatly improve our manuscript. In response to the reviewers' comments, the modifications made to the manuscript have been highlighted in blue.

Reviewer #1:

Comment #1: Please change the name of the protein throughout the manuscript; by NFkB you mean p65..? = NF-κB p65

Response: Yes, the reviewer is correct. We mean p65 By NFκB. The protein is now referred to as “NF-κB p65” instead of “NF-κB” throughout the manuscript (Materials and Methods, Results, Discussion, Figure 4 and Figure 6).

Comment #2: IVIS imaging method description and luciferin/substrate use in mouse and in vitro is lacking. Please add.

Response: We have described the IVIS imaging method and mentioned the details of luciferin substrate in the materials and methods section of the revised manuscript as follows:

“...Tumor formation was confirmed by non-invasive bioluminescence imaging (IVIS Spectrum, In Vivo Imaging System, Caliper Life Sciences) at the Molecular Imaging Facility, ACTREC. Mice were anesthetized by continuous 2% isoflurane inhalation and 100 μl of 30 mg/ml of D-luciferin substrate (D-Luciferin Firefly, potassium salt, Cat No. L-8220, Biosynth Carbosynth) was injected intraperitoneally for bioluminescence imaging (BLI). Living Image 4.5 software was used to measure bioluminescence signal peak using auto exposure mode. For the assessment of metastasis, mice were sacrificed and subjected for BLI imaging after excision of complete tongue with the primary tumor. Further, mice were dissected for *ex-vivo* BLI imaging of internal organs (submaxillary glands and cervical lymph nodes, lungs, liver, kidney, spleen, uterus and ovaries) collected in petri dish and sprayed with D-luciferin substrate to assess distant metastasis using IVIS system...”

Comment #3: Please confirm that you mention in the method description that the animal experiments were performed on female mice only (if this is the case).

Response: All of the *in-vivo* experiments were done with female mice only. We have now mentioned the same in the materials and methods section of the revised manuscript as follows:

“...Cells were injected orthotopically into the tongue of 6-8 week old female nude mice (N=6/group) using needle gauge (30 G) after anaesthetizing the mice with isoflurane...”

Reviewer #2:

Comment #1: The authors did not answer some critical question (Reviewer #2 comments#2). Authors should perform additional experiments to solve this question.

Response: We agree with the reviewer that studying the role of *miR-944/MMP10/AXL*- axis in lymph node metastasis in other stages of tongue cancer is very important and relevant. However, we respectfully submit that this follow up study is outside the scope of the current manuscript.

Comment #2: It is still unclear that authors use the term “early stage” even in the cases have lymph node metastasis. Authors described that “We focused on T1 and T2 stage of tongue cancer since these individuals are known to have occult lymph node metastasis, which reduces their chance of survival

by 50% when compared to those without lymph node metastases.” However, the cases with lymph node metastasis are not early stage....

Response: As per the AJCC (American Joint Committee on Cancer)/ UICC (Union for International Cancer Control) TNM classification system (8th edition), primary oral cavity tumors are clinically staged as T1 (measuring ≤ 2 cm) and T2 (measuring > 2 cm but < 4 cm) based on tumor size, and are classified as early-stage. These individuals were classified as "clinically early-stage" because their lymph node metastasis were not diagnosed on a physical exam or with imaging modalities like PET/CT or ultrasonography. Moreover, the 108 samples used for MMP10 IHC in this study were obtained from a clinical trial, wherein as well these patients were classified as being in the early stages of the disease (D'Cruz *et al.*, *NEJM*, 2015).

However, we do understand the reviewer's concern. Accordingly, per the reviewer's recommendation, we have fully eliminated the term "early-stage" from the revised manuscript at all the 26 different instances (which includes the title, abstract, introduction, results, discussion and Figures 1, 3, 6 of the manuscript).

Comment #3: To demonstrate the role of miR-944/MMP10/AXL axis in early stage of tongue cancer, authors used oral cancer cell lines. AW 13516 and AW 8507 cell lines derived from poorly differentiated SCC and epidermoid carcinoma of the tongue respectively were used in this study. Moreover, CAL27 cells are also used. Are these cell lines “early stage of tongue cancer”?

Response: We agree with the reviewers. The AW13516, AW8507 (Tatake *et al.*, *J. Cancer Res. Clin. Oncol.*, 1990) and CAL27 (Gioanni *et al.*, *Eur. J. Cancer Clin. Oncol.*, 1988) cells used in the study were derived from the advanced stage (T3-T4) tongue cancer patients. In accordance with our previous response to reviewer #2's comment #2, we have removed the phrase "early-stage" from the revised manuscript.

Comment #4: In this study, authors focus on tongue cancer and exclude buccal cancer. It is still unclear why the authors exclude buccal cancer.

Response: As mentioned in response to comment #1 of reviewer #2 (in our previous revision), to avoid the heterogeneity displayed by different oral cavity sub-sites, we focused only on one sub-site (tongue) in this study, which is also the most prevalent subtype among oral cavity cancers in industrialised nations and in younger age group of people in India (Sherin *et al.*, *Indian J. Cancer*, 2008).

Comment #5: In Figure 6, authors added the schematic model (Reviewer#3 comment#7). Downstream of AKT signaling, authors showed mTOR and NF- κ B. However, authors did not demonstrate this.

Response: Per the reviewer's suggestion, we have modified the schematic model (Figure 6) as shown below:

Figure 6: Model depicting miR-944/MMP10/AXL axis to regulate metastasis in tongue cancer. The working model demonstrating the role of MMP10 in promoting metastasis via activation of AXL signaling pathway and is negatively regulated by miR-944 in tongue cancer.

Reviewer #3:

Comment #1: Although the authors provided the quantification and normalization of western blot proteins, the information is demonstrated in the supplementary material. It would be important to provide this information in the main text of the manuscript.

Response: As suggested by the reviewer, we have included the normalized quantitative values of western blot proteins in the revised manuscript figures (Figure 2, 4; Supplementary Figure S2, S3, S7, S10) and mentioned the same in respective figure legends.

Figure 2, A, D, H: Genetic perturbation of MMP10 affects cell proliferation, migration, and invasion of tongue cancer cells. (A) qRT-PCR and immunoblot of MMP10 and GAPDH in AW13516 cells stably overexpressing empty vector or MMP10. Numbers on the blot indicate intensity ratio of MMP10 expression with respect to the vector control lane. (D and H) qRT-PCR and immunoblot of MMP10 and GAPDH in AW8507 (D) and CAL27 (H) cells with sh-NT or stable MMP10 knockdown. Numbers on the blot indicate intensity ratio of MMP10 expression with respect to the sh-NT control lane.

Figure 4, A, B: miRNA-944/MMP10 induced EMT is mediated by the AXL signalling pathway. (A-B) Immunoblot of MMP10, p-AXL, AXL, p-AKT, AKT, p-mTOR, mTOR, p-NF-kB p65, NF-kB p65, Vinculin and β-actin upon overexpression/knockdown of MMP10 or overexpression/knockdown of AXL in tongue cancer cell lines. Vinculin or β-actin was used as loading control. Numbers on the blot indicate intensity ratio of target protein expression with respect to the vector control lane.

Supplementary Figure S2C: Expression of MMP10 in tongue cancer cell lines. (C) MMP10 expression at the protein level in tongue cancer cell lines was measured by Western blotting assay. Vinculin was used as loading control. Numbers on the blot indicate intensity ratio for MMP10, normalized to Vinculin levels in the respective cell lines. Data shown are representative of three independent experiments.

Supplementary Figure S3, A, E: Knockdown of MMP10 does not affect apoptosis of tongue cancer cells. (A and E) Bar plot of cell viability assay performed using Propidium Iodide (PI) staining indicates the percentage of late apoptotic and necrotic cells in MMP10 knockdown clones (sh-1, sh-2 and sh-3) and non-targeting shRNA control clones (sh-NT) of AW8507 (A) and CAL27 (E). (Bottom Western blots) Western blots indicate the expression of Caspase 3 and PARP cleavage in the MMP10 knockdown and vector control cells. Numbers on the blot indicate intensity ratio of target protein expression with respect to the vector control lane. β -actin or Tubulin was used as loading control.

Supplementary Figure S7D: Differential expression analysis and validation of deregulated genes and EMT markers upon overexpression of MMP10 in AW13516 cell line. (D) Western blot analysis of E-cadherin, Vimentin, MMP9 and β -catenin upon overexpression of MMP10 in AW13516. Numbers on the blot indicate intensity ratio of target protein expression in MMP10 OE clones with respect to the vector control lane. Vinculin was used as reference control.

Comment #2: The authors should provide the method employed for the normalization and quantification of western blot proteins.

Response: We have detailed the method used for normalization and quantification of western blot proteins in the materials and methods section of the revised manuscript as follows:

“...Western blot protein intensities were normalized and quantified using ImageJ software. TIFF files of western blot images were used for quantification. Using the rectangular tool in ImageJ, protein band intensity was measured and normalized with the background intensity in same blot to obtain the normalized band intensity. The actual band intensities were obtained by further normalization with the reference loading control. Fold change expression of each protein was obtained by comparing the actual band intensity of the protein in the overexpression/ knockdown group versus vector control group...”

Reviewer #4:

Comment #1: Rev4, comment/question 8.1. (patient samples for immunohistochemistry, Fig1A): Retrospectively collected or not, buccal mucosal samples are not OTSCC and thus should A) either not be included in the analyses (especially since their exclusion does not drastically change the results of the analyses) OR B) the figure 1 legend should be modified to include information of the buccal mucosal samples included in the analyses (number of such samples in each group, title of the figure etc.).

Response: We agree with the reviewer. Per the reviewer's advice, we have modified the figure 1 title and legend to include the information of the buccal mucosa cancer samples used in the analysis, as shown below:

“...**Figure 1: MMP10 overexpression is associated with nodal metastasis in tongue cancer (N=208) and buccal mucosa cancer (N=11) patients.** (A) Immunohistochemistry (IHC) of MMP10 in primary tongue tumor (N=98) and buccal mucosa tumor samples (N=10). One dot represents IHC score of MMP10 in one sample. “Node Pos” and “Node Neg” represent primary tumor sample collected from patients with and without lymph node metastasis, respectively. (B) Quantitative real-time PCR (qRT-PCR) of *MMP10* transcript expression in primary tongue tumor (N=110) and buccal mucosa tumor samples (N=1). *GAPDH* was used as reference control. Data is plotted as box plot representation of delta Ct (dCT) values...”

Comment #2: Rev4, comment/question 15 (TCGA): It is evident why the authors want to emphasize the survival in HNSCC dataset instead of OTSCC dataset. Yet for transparency, it should be mentioned in either the results or discussion section that similar conclusion cannot be drawn from the TCGA-OTSCC data alone. The reasons for this discrepancy can be mentioned as well, e.g. small number of samples etc.

Response: We agree with the reviewer. Accordingly, we have mentioned the survival analysis performed using TCGA-OTSCC data based on *MMP10* or *AXL* expression in the results section as follows:

“...Survival analysis suggests high expression of *MMP10* or *AXL* was significantly ($p<0.05$) associated with poor overall survival of head and neck squamous cell carcinoma (HNSCC) patients (Supplementary Figure S9), but not just tongue cancer patients (data not shown), which could be because there were fewer tongue cancer patients...”

Comment #3: Please consider updating the information in the Reporting Summary. Information on Flow cytometry analysis is missing and potentially also of use of clinical data (unless the reporting summary specifically addresses only clinical trials).

Response: We apologize for the omission on our part, and thank the reviewer to bring this to our attention. Following the reviewer's suggestion, we have updated the flow cytometry analysis information in the reporting summary.

Furthermore, as correctly pointed by the reviewer, the "clinical data" field in the "Materials & experimental systems" section of the reporting summary specifically refers to the clinical trials, hence we did not include this under the "Materials & experimental systems" section. However, we have detailed the clinical data used in this study under the "Human research participants" section of the reporting summary.

REVIEWERS' COMMENTS:

Reviewer #1 (Remarks to the Author):

The authors have addressed all my concerns. I have no further questions or comments on the manuscript.

The authors have substantially improved the manuscript according to reviewers' suggestions on revisions.

Reviewer #2 (Remarks to the Author):

I think that the authors satisfactory responded to the comments from the reviewers. Therefore, I can recommend this paper for publication.

Reviewer #3 (Remarks to the Author):

As requested in the previous round of revision, the authors provided the quantification and normalization of western blot proteins for all figures in the manuscript. Furthermore, they described in the details the method used for normalization and quantification of western blot proteins.

Reviewer #4 (Remarks to the Author):

This review concerns the second revision of manuscript titled "Role of miR-944/MMP10/AXL- axis in Lymph Node Metastasis in Tongue Cancer" by Dharavat et al.

The authors have made sufficient additional modifications to the the manuscript as suggested by the reviewers, which has improved the manuscript even further.

There is only one discrepancy to be addressed: The manuscript heavily focuses on tongue cancer and the authors also claim this in their response to the Reviewer 2's comment 4. Yet the buccal mucosa samples still remain in the IHC and qRT-PCR analyses (as shown in Figure 1 and Table 1). This contradicts what authors claim themselves. For clarity and to omit discrepancy within the manuscript/analyses, the authors are strongly advised to leave out these (only) 11 buccal tumor samples from the analyses. Especially since they do not affect the outcome of the analyses presented in Figure 1 (as stated by the authors previously in earlier revision, Reviewer 4 Comment 8.1).